# Disentangling Voice and Content with Self-Supervision for Speaker Recognition

**Tianchi Liu[1,2], Kong Aik Lee**[*3, 1], **Qiongqiong Wang[1], Haizhou Li**[4, 2]

[1] Institute for Infocomm Research (I[2]R), Agency for Science, Technology and Research (A[*]STAR), Singapore
[2] Dept. of Electrical and Computer Engineering, National University of Singapore, Singapore
[3] Dept. of Electrical and Electronic Engineering, Hong Kong Polytechnic University, Hong Kong
[4] School of Data Science, The Chinese University of Hong Kong, Shenzhen, China

{liu_tianchi, wang_qiongqiong}@i2r.a-star.edu.sg, kongaik.lee@ieee.org, haizhouli@cuhk.edu.cn

## Abstract

For speaker recognition, it is difficult to extract an accurate speaker representation from speech because of its mixture of speaker traits and content. This paper proposes a disentanglement framework that simultaneously models speaker traits and content variability in speech. It is realized with the use of three Gaussian inference layers, each consisting of a learnable transition model that extracts distinct speech components. Notably, a strengthened transition model is specifically designed to model complex speech dynamics. We also propose a self-supervision method to dynamically disentangle content without the use of labels other than speaker identities. The efficacy of the proposed framework is validated via experiments conducted on the VoxCeleb and SITW datasets with 9.56% and 8.24% average reductions in EER and minDCF, respectively. Since neither additional model training nor data is specifically needed, it is easily applicable in practical use.

## 1 Introduction

Automatic speaker recognition aims to identify a person from his/her voice [2] based on speech recordings [22]. Typically, two fixed-dimensional representations are extracted from the enrollment and test speech utterances, respectively [68]. Then the recognition procedure is done by measuring their similarity [22]. These representations are referred to as the speaker embeddings [67]. The concept of speaker embedding is similar to that of the face embedding [64] in the face recognition task and the token embedding [16] in the language model, while the main difference lies in the carriers of the information source and the nature of downstream tasks. Different from the discrete sequential inputs in Natural Language Processing (NLP) and continuous inputs in Computer Vision (CV), speech signals are continuous-valued variable-length sequences [24]. Speech signals contain both speaker traits and content [32]. For speaker recognition, to form a refined speaker embedding of a speech signal, conventional methods aggregate the frame-level features by pooling across the time-axis to extract speaker characteristics while factoring out content information. Simple temporal aggregation fails to disentangle the content information and therefore affects the quality of the resulting embeddings. The content information is regarded as unwanted variations hindering the accurate representation of voice characteristics.

To reduce the effects of the content variation, prior works model the phonetic content representation and use it as a reference for speaker embedding extraction. In [89], the phonetic bottleneck features from the last hidden layer of a pre-trained automatic speech recognition (ASR) network are combined with raw acoustic features to normalize the phonetic variations. A coupled stem is designed in [88] to jointly learn acoustic features and frame-level ASR bottleneck features. In [42], a phonetic attention mask dynamically generated from a sub-branch is used to benefit speaker recognition. The existing methods can be summarized into two types in terms of content representation modeling: (1) by a

---

[*]Corresponding author.

37th Conference on Neural Information Processing Systems (NeurIPS 2023).

pre-trained ASR model [57, 34, 45, 89, 46], and (2) by a jointly trained multi-task model with extra components for content representations [43, 88, 17, 46, 80, 44, 42]. These methods prove that the utilization of content representation benefits speaker recognition. However, both types lead to an obvious limit in practical applications. The employment of a pre-trained ASR model greatly increases the parameters and computation complexity in the inference, as the ASR models are typically one or two orders of magnitude larger than the speaker recognition model [87, 71]. The joint training of the speaker recognition model and extra components for content representations requires either an extra dataset with text labels in addition to the dataset with speaker identities, or one dataset with speaker identities and text labels simultaneously which is expensive and not easy to come by. For the former type of joint training, additional components are still needed with extra effort and may encounter optimization difficulties.

For the purpose of speaker recognition, one aims to derive an embedding vector representing the vocal characteristics of the speaker. The benefits of leveraging content representation are significant, while the drawbacks of text labels and extra model requirements are obvious. Motivated by this, we seek to design a novel framework to solve these problems.

Speech signals consist of many components, of which the two major parts are speaker traits and speech content [32, 14]. In this paper, we focus on decomposing speech signals into static and dynamic components. The former is static, i.e., fixed, with respect to temporal evolution and dominated by speaker characteristics, while the latter mainly consists of speech content with some other components, such as the prosody. Based on this assumption, we design a framework with three Gaussian inference layers which aims to decompose the static and dynamic components of the speech. The static part is modeled by static Gaussian inference with the criterion of speaker classification loss. During inference, the static latent factor is associated with the vocal characteristics of the speaker, and we refer to it as speaker representation. The remaining dynamic components related to verbal content are modeled by dynamic Gaussian inference. The motivation of the three-layer design is simple and intuitive – the speaker representation from *layer 1* is not accurate and may include content information as the content-related reference is absent in training. However, it helps *layer 2* extract the content representation which can be used as the reference for *layer 3*. Thus, a more accurate speaker representation is extracted in *layer 3*. It is worth noting that the framework is trained without text labels and no extra model or branch is employed thus not much increase in model size. This is achieved with the guide of self-supervision and considered content representations. We named this framework as **Rec**urrent **Xi**-vector (RecXi). The major contributions are summarized as follows:

- **RecXi: A novel disentangling framework** with the following features: (1) We enhance the xi-vector [33] with the ability to capture temporal information and name it the recurrent xi-vector layer. (2) A frame-wise content-aware transition model $\mathbf{G}_t$ is proposed for modeling dynamic components of the speech. (3) A novel design of three layers of Gaussian inference is proposed to disentangle speaker and content representations by static and dynamic modeling with the corresponding counterpart removal, respectively.
- **A self-supervision method** is proposed to guide content disentanglement by preserving speaker traits. It reduces the impact of the absence of text labels.

## 2 Related Work

**Conventional Speaker Recognition Methods.** Speaker embeddings are fixed-length continuous-value vectors extracted from variable-length utterances to represent speaker characteristics. They live in a simpler Euclidean space in which distance can be measured for the comparison between speakers [78] . A well-known example of the extractor is the x-vector framework [67], which mainly consisted of the following three components: **1). An encoder** is implemented by stacking multiple layers of time-delay neural network (TDNN) and used to extract the frame-level features from utterances [58]. Recent works strengthen the encoder by replacing the simple TDNNs with powerful ECAPA-TDNN [15] and its variants [41, 74, 23, 53, 47], or ResNet series models [90, 36, 84, 83, 51]. **2). A temporal aggregation layer** aggregates frame-level features from the encoder into fixed-length utterance-level representations. **3). A decoder** classifies the utterance-level representations to speaker classes for supervised learning by a classification loss. The decoder stacks several fully-connected layers including one bottleneck layer used to extract speaker embedding.

**Speech Disentanglement.** Various informational factors are carried by the speech signal, and the speech disentanglement ultimately depends on which informational factors are desired and how they

will be used [81]. For speaker recognition, in addition to the use of content information we introduced above, some works attempt to disentangle speaker representation with the removal of irrelevant information, like devices, noise, and channel information with corresponding labels [50, 52, 28]. [52] minimizes the mutual information between speaker and device embeddings, with the goal of reducing their interdependence. In [40], nuisance variables like gender and accent are removed from speaker embeddings by learning two separate orthogonal representations. Many other speech-relevant tasks also show great improvements by disentangling these two components properly. For speaker diarization, the ASR model is employed to obtain content representations leveraged by speaker embedding extractions [70, 29]. In voice conversion or personalized voice generation tasks, a popular method is to disentangle the speech into linguistic and speaker representations before performing the generation [26, 86, 92, 11, 18, 91, 38]. In ASR, speaker information is extracted for speaker variants removal for performance improvements [49, 35, 25] or privacy preserving [69]. Similarly, auxiliary network-based speaker adaptation [63] or residual adapter [75] are explored to handle large variations in the speech signals produced by different individuals. These works have led us to the design of RecXi, which is the first speaker and content disentanglement framework for speaker recognition in the absence of extra labels for practical use to the best of our knowledge.

**Self-Supervision in Speech.** Self-supervised learning has been the dominant approach for utilizing unlabeled data with impressive success [1, 16, 7, 20]. In speaker recognition, *contrastive learning* [9] is used to force the encoder to produce similar activation between a positive pair. The positive pair can be two disjoint segments from the same utterance [6, 85] or from cross-referencing between speech and face data [73]. The trained speaker encoder also can produce pseudo speaker labels for supervised learning [72, 5]. In target speaker extraction, self-supervision is helpful to find the speech-lip synchronization [56]. In voice conversion, much research focuses on applying *variational auto-encoder* (VAE) to disentangle the speaker [10, 27, 59]. Similar to reconstruction-based methods, mutual information (MI) and identity change loss are used in [30] for robust speaker disentanglement. ContentVec [60] disentangles and removes speaker variations from the speech representation model HuBERT [24] for various downstream tasks, such as language modeling and zero-shot content probe.

Considering the main objective of this work is to benefit speaker recognition by disentangling the speaker with the speaker classification supervision, we want to clarify that the content disentanglement and corresponding self-supervision serve this final target. Therefore, the self-supervision method is designed to be simple yet effective while avoiding extra signal re-constructors or training efforts.

## 3 Approach

### 3.1 Xi-vector

Many works have been proposed to estimate the uncertainty for the speaker embedding [65, 33, 78]. Among them, the xi-vector [33] is proposed to enhance x-vector embedding in handling the uncertainty caused by the random intrinsic and extrinsic variations of human voices. Specifically, an input frame $\mathbf{x}_t$ is encoded to a point-wise estimate $\mathbf{z}_t$ in a latent space by an encoder. And an auxiliary network is employed to characterize the frame-level uncertainty with a posterior covariance matrix $\mathbf{L}_t^{-1}$ associated with $\mathbf{z}_t$ as shown in the encoder part of Figure 1. These two operations are formulated as

$$\mathbf{z}_t = f_{\text{enc}}(\mathbf{x}_t | \mathbf{x}_t^{t \pm n}), \qquad (1) \qquad \log \mathbf{L}_t = g_{\text{enc}}(\mathbf{x}_t | \mathbf{x}_t^{t \pm n}), \qquad (2)$$

where the neighbour frames $\mathbf{x}_t^{t \pm n}$ of time $t$ are taken into consideration for the frame-wise estimation. The precision matrices $\mathbf{L}_t$ are assumed to be diagonal and are estimated as log-precision for the convenience of following steps. It is assumed that a *linear Gaussian model* is responsible to generate the representations $\mathbf{z}_t$ as follows:

$$\text{generative model}: \mathbf{z}_t = \mathbf{h} + \epsilon_t, \text{latent variable}: \mathbf{h} \sim \mathcal{N}(\phi_{\text{p}}, \mathbf{P}_{\text{p}}^{-1}), \text{uncertainty}: \epsilon_t \sim \mathcal{N}(\mathbf{0}, \mathbf{L}_t^{-1}). \quad (3)$$

Here, $\mathbf{h}$ is the latent variable for the entire utterance, and $\epsilon_t$ is a frame-wise random variable for the uncertainty covariance measure, $\phi_{\text{p}}$ and $\mathbf{P}_{\text{p}}^{-1}$ are the Gaussian prior mean and covariance matrix of the variable $\mathbf{h}$. The posterior mean vector $\phi_{\text{s}}$ and precision matrix $\mathbf{P}_{\text{s}}$ for the whole sequence are formulated as

$$\phi_{\text{s}} = \mathbf{P}_{\text{s}}^{-1} \left[ \sum_{t=1}^{T} \mathbf{L}_t \mathbf{z}_t + \mathbf{P}_{\text{p}} \phi_{\text{p}} \right], \qquad (4) \qquad \mathbf{P}_{\text{s}} = \sum_{t=1}^{T} \mathbf{L}_t + \mathbf{P}_{\text{p}}, \qquad (5)$$

where the posterior mean $\phi_{\text{s}}$ is enriched by frame-wise uncertainty and is further used to be decoded into speaker embedding.

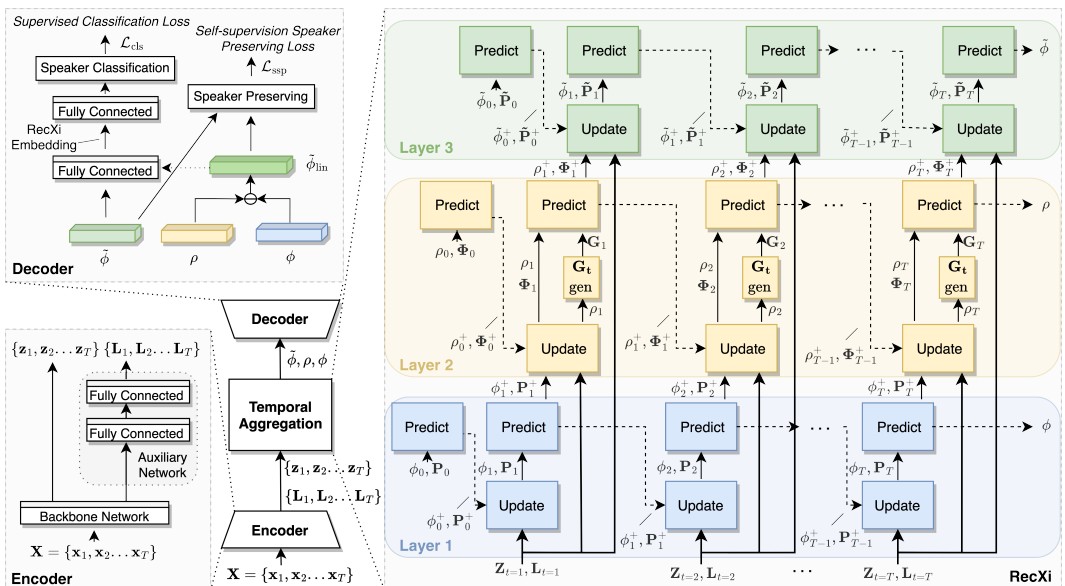

Figure 1: The network architecture of the proposed RecXi system with self-supervision. The structure at the bottom-middle of the figure is a simplified speaker recognition system. The figures in three dotted boxes are the specific explanations of its three parts. At the encoder, $\{\mathbf{x}_1, \mathbf{x}_2...\mathbf{x}_T\}$ is the input sequence of length $T$. The three colors of blue, orange, and green indicate three recursive layers. For each layer, the inference flow for frames $\mathbf{x}_1$, $\mathbf{x}_2$ and $\mathbf{x}_T$ is drawn, while the frames in between are replaced by '...'. The dashed lines indicate recurrent operation and the solid lines represent the operations within the same frame. The block with $\mathbf{G}_t$ gen indicates the filter generator. For the decoder, $\ominus$ is a subtraction operation. The dotted line indicates the operation is optional.

## 3.2 Disentangling Speaker and Content Representations

**Basic Recurrent Xi-vector Layer.** Xi-vector has the advantage of modeling static components of the speech by the utterance-level speaker representation aggregation with the use of uncertainty estimation, while the drawback of modeling dynamic signals is obvious. For approximating non-linear functions in high-dimensional feature spaces and restricting the inference through the adjacent Gaussian hidden state efficiently, similar to [4], a learnable linear transition model $\mathbf{G}$ is applied. We further extend the xi-vector into a recursive form with frame-level estimation. It is worth noting that xi-vector is a special case of the basic recurrent xi-vector when $\mathbf{G}$ is an identity matrix.

The Gaussian inference in the latent space can be implemented in two stages: *predict stage* and *update stage*. In the *predict stage*, the predictive mean vector $\rho_t^+$ and precision matrix $\mathbf{\Phi}_t^+$ are formulated as

$$\rho_t^+ = \mathbf{G}\rho_t, \qquad (6) \qquad\qquad \mathbf{\Phi}_t^+ = [\mathbf{G}\mathbf{\Phi}_t^{-1}\mathbf{G}^{\mathrm{T}}]^{-1}, \qquad (7)$$

where $\mathbf{G}$ represents a linear transition model, and $^+$ indicates the results of the given frame $t$ after the *predict stage*. T indicates a transpose operation.

For sake of clarity, we assign the time (i.e., frame) index $t = 0$ to the priors, such that $\rho_0 = \rho_{\mathrm{p}}$, $\mathbf{\Phi}_0 = \mathbf{\Phi}_{\mathrm{p}}$. In the *update stage*, the posterior mean vector $\rho_t$ of frame $t$ is derived by incorporating the previously predicted posterior mean and precision $\{\rho_{t-1}^+, \mathbf{\Phi}_{t-1}^+\}$ of the hidden states of the former frame $t-1$ with encoded features $\mathbf{z}_t$ and estimated uncertainty $\mathbf{L}_t$ of current frame $t$. The uncertainty measures $\mathbf{L}_t$ of frame $t$ is added into the posterior precision matrix. These two operations are derived as

$$\rho_t = \mathbf{\Phi}_t^{-1}[\mathbf{L}_t\mathbf{z}_t + \mathbf{\Phi}_{t-1}^+\rho_{t-1}^+], \qquad (8) \qquad\qquad \mathbf{\Phi}_t = \mathbf{L}_t + \mathbf{\Phi}_{t-1}^+. \qquad (9)$$

**Frame-wise Content-aware Transition Model $\mathbf{G}_t$.** Speech signals are a complex mixture of dynamic information sources. Even though the transition model $\mathbf{G}$ is learnable during training, it remains the same across all the hidden states of Gaussian inference for each sample. To enhance the ability to model dynamic speech components for content disentanglement, we propose a transition

model $\mathbf{G}_t$ which is dynamically adjusted by a filter generator for each frame according to the content during the Gaussian inference and hereby named frame-wise content-aware transition model.

Specifically, a set of $N$ learnable transition matrices $\{\mathbf{G}'_1, \mathbf{G}'_2...\mathbf{G}'_N\}$ is employed to model $N$ dynamic components in speech signals. For each frame $t$, a vector $\mathbf{w}_t \in \mathbb{R}^N$ is generated representing the importance of different dynamic components to content representation modeling, by observing content information $\rho_t$ from *update stage*. The content-aware transition model for frame $t$ is finally obtained as a weighted sum over the $N$ component-dependent transition models with weight $w_{t,n}$.

$$\mathbf{G}_t = \sum_{n=1}^{N} w_{t,n} \mathbf{G}'_n, \quad (10) \qquad w_{t,n} = [\mathbf{w}_t]_n, \quad (11) \qquad \mathbf{w}_t = \sigma\left(f\left(\rho_t\right)\right), \quad (12)$$

where $\mathbf{w}_t \in \mathbb{R}^N$. $\sigma$ and $f$ indicate the $Softmax$ function and a non-linear operation, respectively. In this work, the non-linear operation is designed as two fully connected layers with a ReLU [19] activation function in between. It is to be noted that this procedure is frame-wise as the dynamic components vary along the sequence of $T$. As an extension to Equations (6) and (7), the *predict stage* of recurrent xi-vector is reformulated as

$$\rho_t^+ = \mathbf{G}_t \rho_t, \quad (13) \qquad\qquad \Phi_t^+ = [\mathbf{G}_t \Phi_t^{-1} \mathbf{G}_t^{\mathrm{T}}]^{-1}. \quad (14)$$

**Three Layers of Gaussian Inference.** We propose a structure based on three layers of Gaussian inference for disentangling speaker and content representations in the absence of text labels. In this work, this structure is utilized in the temporal aggregation layer and named RecXi. The three layers of Gaussian inferences each aim at precursor speaker representation, disentangled content representation, and disentangled speaker representation, as shown in the RecXi part of Figure 1. The last frame's representations from each layer are used for deriving embeddings within the decoder. The details of each layer are discussed as follows.

*Layer 1*: **Precursor Speaker Representation.** This is a basic recurrent xi-vector layer that aims to represent the speaker characteristics. Therefore, the transition model of the Gaussian inference is set as an identity matrix to model static components of the speech. The *predict stage* can be derived from Equations (6) and (7) as

$$\phi_t^+ = \phi_t, \quad (15) \qquad\qquad \mathbf{P}_t^+ = \mathbf{P}_t. \quad (16)$$

The speaker representation from this layer is similar to that in the original xi-vector, where the content information from the high-dimensional frame-level features remains and affects the speaker embedding quality. We refer to the representation of this layer as the precursor speaker representation. Specifically, the frame-wise representations of posterior mean $\phi_t$ and precision $\mathbf{P}_t$ of the hidden state $\mathbf{h}$ of frame $t$ are estimated. They are derived partly according to frame-level features $\mathbf{z_t}$ and uncertainty $\mathbf{L_t}$ extracted from the corresponding frame $t$, and partly from previous frames $\{1, 2, ...t-1\}$ by recursively passing in the posteriors from the previous frame. The *update stage* is formulated as

$$\phi_t = \mathbf{P}_t^{-1}[\mathbf{L}_t \mathbf{z}_t + \mathbf{P}_{t-1}^+ \phi_{t-1}^+], \quad (17) \qquad\qquad \mathbf{P}_t = \mathbf{L}_t + \mathbf{P}_{t-1}^+. \quad (18)$$

*Layer 2*: **Disentangled Content Representation.** This layer aims to disentangle content representation from the sequence. To model dynamic subtle content changes, the frame-wise content-aware transition model $\mathbf{G}_t$ introduced above is applied. As shown in the orange part of Figure 1, the transition model $\mathbf{G}_t$ is generated by a filter generator according to Equations (10), (11) and (12). Equations (13) and (14) are applied to the *predict stage* of *layer 2*.

To disentangle the content representation more effectively, in addition to equipping the layer with dynamic modeling ability, we further attempt to remove speaker information from the frame-level features for each frame during Gaussian inference. This ensures that the remaining information is more likely to be dynamic and associated with the content.

The posterior mean $\phi^+$ and posterior precision $\mathbf{P}^+$ of the hidden state of *layer 1* are rich with speaker information and utilized by the *update stage* of *layer 2*. Benefiting from the linear operations in this three layers design, the speaker removal operation can simply be done by subtracting the posterior mean $\phi^+$ from features $\mathbf{z}$ while adding the posterior covariance matrix $(\mathbf{P}^+)^{-1}$ into the uncertainty estimation matrix $\mathbf{L}^{-1}$. The procedure is dynamically processed for each frame, and the *update stage* is formulated as

$$\rho_t = \mathbf{\Phi}_t^{-1}[(\mathbf{L}_t^{-1} + (\mathbf{P}_t^+)^{-1})^{-1}(\mathbf{z}_t - \phi_t^+) + \mathbf{\Phi}_{t-1}^+ \rho_{t-1}^+], \tag{19}$$

$$\mathbf{\Phi}_t = (\mathbf{L}_t^{-1} + (\mathbf{P}_t^+)^{-1})^{-1} + \mathbf{\Phi}_{t-1}^+. \tag{20}$$

*Layer 3*: **Disentangled Speaker Representation.** This layer uses the Gaussian inference with an identity matrix for modeling static components of the speech. Furthermore, the speaker classification loss is applied to the output of this layer and provides supervision restrictions. Different from *layer 1*, *layer 3* is designed to model the desired disentangled speaker representation by removing the content information provided by *layer 2* from the frame-level features. The procedure is similar to that in *layer 2*, and the *predict stage* is formulated as

$$\tilde{\phi}_t^+ = \tilde{\phi}_t, \qquad (21) \qquad\qquad \tilde{\mathbf{P}}_t^+ = \tilde{\mathbf{P}}_t, \qquad (22)$$

where $\tilde{\phi}_t^+$ and $\tilde{\mathbf{P}}_t^+$ are the posterior mean and posterior precision , respectively. The symbol of $\tilde{\ }$ is used to differentiate them from the posteriors in *layer 1*. The *update stage* is formulated as

$$\tilde{\phi}_t = \tilde{\mathbf{P}}_t^{-1}[(\mathbf{L}_t^{-1} + (\mathbf{\Phi}_t^+)^{-1})^{-1}(\mathbf{z}_t - \rho_t^+) + \tilde{\mathbf{P}}_{t-1}^+ \tilde{\phi}_{t-1}^+], \tag{23}$$

$$\tilde{\mathbf{P}}_t = (\mathbf{L}_t^{-1} + (\mathbf{\Phi}_t^+)^{-1})^{-1} + \tilde{\mathbf{P}}_{t-1}^+. \tag{24}$$

It is worth noting that to avoid expensive matrix multiplication operations and numerically problematic matrix inversions, a simplified implementation is adopted (see Appendix A).

## 3.3 Speech Disentanglement with Self-supervision

A well-trained speaker embedding neural network requires a huge number of speakers and utterances to achieve discriminative ability. For such a large dataset, text labels are very difficult to come by. In the absence of text labels, disentangling content information from the speech is a very difficult task.

In Section 3.2, the *layer 3* of RecXi is designed to be the desired speaker representation. This is achieved by static Gaussian inference with the assumption that disentangled content representation provided by *layer 2* is reliable. We note that the disentangled speaker representation from *layer 3* is directly optimized through the classification criterion, while the optimization for disentangling content information is indirect through content removal operations in Gaussian inference. In order to ameliorate the shortcoming due to the absence of text label and to provide an extra supervisor for *layer 2*, we propose a self-supervision method to preserve speaker information via knowledge distillation in a similar fashion to those proposed in [62, 76, 8] for the teacher-student pair.

Generally speaking, large models usually outperform small models, while the small model is computationally cheaper [76]. Knowledge distillation aims to benefit a small model with the guidance of a large model. Different from the general knowledge distillation methods, our 'teacher' and 'student' are two different layers within the RecXi. Since this guidance comes from the RecXi itself, and all the training data is considered as unlabelled data for the content disentanglement task, the proposed method is considered a self-supervision method [3].

The output $\phi$ from *layer 1* of RecXi is precursor speaker representation, and the content information remains, while *layer 2* is designed to disentangle content representation $\rho$. Benefiting from the linear operations used in RecXi, we can remove content information and preserve speaker representation by subtracting the content representation $\rho$ of *layer 2* from the precursor speaker representation $\phi$ of *layer 1*. This speaker representation is marked as $\tilde{\phi}_{\mathrm{lin}}$ and derived as

$$\tilde{\phi}_{\mathrm{lin}} = \phi - \rho \tag{25}$$

where the 'lin' indicates that it is obtained by a **lin**ear operation, differing from $\tilde{\phi}$ in *layer 3* which is disentangled by Gaussian inference. For the same input sequence, the speaker representation $\tilde{\phi}_{\mathrm{lin}}$ derived by this linear operation should be consistent with $\tilde{\phi}$ obtained from disentanglement. Therefore, by restricting their difference, the gradient from supervision speaker classification loss will form a guide for *layer 2* through the linear operation in Equation (25). This serves as an extra supervisory signal different from the constraints imposed by speaker removal operation during the Gaussian inference. As $\tilde{\phi}$ is optimized by classification loss directly, it is considered as a 'teacher',

while $\tilde{\phi}_{\text{lin}}$ is a 'student'. Since $\tilde{\phi}_{\text{lin}}$ is derived by preserving speaker representation in Equation (25), we name the loss as **s**elf-supervision **s**peaker **p**reserving loss $\mathcal{L}_{\text{ssp}}$.

The $\mathcal{L}_{\text{ssp}}$ loss can be generated by different knowledge distillation methods, a simple comparison is available in Appendix D. In this work, the idea of similarity-preserving loss [76] is in line with our layer-wise design and is inherited to guide the student towards the activation correlations induced in the teacher, instead of mimicking the teacher's representation space directly. It is derived as

$$\mathcal{L}_{\text{ssp}} = \frac{1}{b^2} \sum_{(s,s') \in \kappa} \left\| \left( \left\| \tilde{\phi}^{(s)} \tilde{\phi}^{(s)\text{T}} \right\|_2 - \left\| \tilde{\phi}_{\text{lin}}^{(s')} \tilde{\phi}_{\text{lin}}^{(s')\text{T}} \right\|_2 \right) \right\|_F^2, \tag{26}$$

where $\kappa$ collects all the $(s, s')$ pairs from the same mini-batch. $\|\cdot\|_2$ and $\|\cdot\|_F$ indicate the row-wise L2 normalization and Frobenius norm, respectively. T indicates a transpose operation and $b$ indicates the batch size. We define the total loss for training the framework as

$$\mathcal{L}_{\text{total}} = \alpha \mathcal{L}_{\text{cls}} + \beta \mathcal{L}_{\text{ssp}}, \tag{27}$$

where $\mathcal{L}_{cls}$ is the speaker classification loss, $\alpha$ and $\beta$ are hyperparameters for balancing the total loss.

## 4   Experiments Setup

### 4.1   Dataset, Training Strategy, and Evaluation Protocol

The experiments are conducted on VoxCeleb1 [54], VoxCeleb2 [13], and the Speaker in the Wild (SITW) [48] datasets. It is worth noting that the text labels are not available for these datasets. For a fair comparison, the baselines are all re-implemented and trained with the same strategy as RecXi which follows that in ECAPA-TDNN [15]. All the models are evaluated by the performance in terms of equal error rate (EER) and the minimum detection cost function (minDCF). Detailed descriptions of datasets, training strategy, and evaluation protocol are available in Appendix C.

### 4.2   Systems Description

We evaluate the systems that are combinations of different backbone models in the encoder and different aggregation layers. To verify the compatibility of the proposed RecXi, both 2D convolution (Conv2D)-based and time delay neural network (TDNN)-based backbones are adopted:

- **1) ECAPA-TDNN** [15] is a state-of-the-art speaker recognition model based on TDNNs. The layers before the temporal aggregation layer are considered the backbone network.

- **2) ResNet [21] and tResNet** represent the models based on 2D convolution. The modified ResNet34 [77] is adopted as a baseline. In order to model more local regions with larger frequency bandwidths at different scales, we further modify the ResNet34 in [77] by simply changing the stride strategy and name it tResNet. The details are provided in Appendix B.

The following are temporal aggregation layers. The first three are considered baselines:

- **1) Temporal Statistics Pooling** (term as TSP from here onwards) [66] is a well-known aggregation method. It is the default option for x-vector and adopted in ResNet for speaker recognition [12, 90].

- **2) Channel- and Context-dependent Statistics Pooling** (term as Chan.&Con. from here onwards) [15] is the default aggregation layer in ECAPA-TDNN, modified from the attentive statistics pooling [55] by adding channel-dependent frame attention and allowing the self-attention produced across global properties.

- **3) Xi-vector Posterior Inference** (term as Xi from here onwards) [33] inserts uncertainty estimation into speaker embeddings as introduced in Section 3.1.

- **4) RecXi ($\tilde{\phi}$) and RecXi ($\tilde{\phi}$, $\tilde{\phi}_{\text{lin}}$)** are the proposed methods. The former uses only $\tilde{\phi}$ as the input to the decoder to derive the speaker embedding, and the latter uses a concatenation of both $\tilde{\phi}$ and $\tilde{\phi}_{\text{lin}}$ as the input.

Table 1: Performance in EER(%) and minDCF of the state-of-the-art (SOTA) systems and proposed RecXi on VoxCeleb1 and SITW test sets. Aug. indicates whether the system is trained with data augmentations. Para. is the number of parameters in million. Systems with † are re-implemented. The systems with ‡ following the format of 'aggregation layer + backbone model' are baselines. # is the index number for the implemented systems.

| # | System | Aug. | Para. | VoxCeleb1-O | | VoxCeleb1-H | | VoxCeleb1-E | | SITW *eval* | |
|---|---|---|---|---|---|---|---|---|---|---|---|
| | | | | EER | minDCF | EER | minDCF | EER | minDCF | EER | minDCF |
| - | ResNet34-SKDFE [39] | ✗ | 5.98 | 1.44 | 0.168 | 2.76 | 0.278 | 1.59 | 0.179 | - | - |
| - | H/ASP(AAM-softmax) [31] | ✗ | 8.0 | 1.25 | - | 2.78 | - | 1.56 | - | - | - |
| - | H/ASP(AP+softmax) [31] | ✗ | 8.0 | 1.21 | - | 2.77 | - | 1.42 | - | - | - |
| - | SI-Net50 [37] | ✗ | - | 1.28 | - | 2.50 | - | 1.28 | - | 2.54 | - |
| - | RSKNet-MTSP [82] | ✗ | 13.9 | **1.05** | 0.159 | 2.52 | 0.237 | 1.30 | 0.152 | 2.27 | 0.228 |
| 1 | ECAPA-TDNN [15]† | ✗ | 6.19 | 1.377 | 0.137 | 2.674 | 0.235 | 1.451 | 0.153 | 2.542 | 0.208 |
| 2 | ResNet34 [77]† | ✗ | 6.63 | 1.489 | 0.155 | 2.500 | 0.224 | 1.423 | 0.158 | 2.378 | 0.208 |
| 3 | TSP + **proposed tResNet**‡ | ✗ | 6.21 | 1.396 | 0.135 | 2.257 | 0.204 | 1.281 | 0.141 | 2.250 | 0.200 |
| 4 | Xi [33] + **proposed tResNet**‡ | ✗ | 6.54 | 1.295 | **0.120** | 2.169 | 0.202 | 1.224 | 0.142 | 2.150 | 0.189 |
| 5 | **RecXi**($\tilde{\phi}, \tilde{\phi}_{\text{lin}}$) with $\mathcal{L}_{\text{ssp}}$ + tResNet | ✗ | 7.06 | 1.196 | 0.122 | **2.097** | **0.196** | **1.197** | **0.124** | **1.832** | **0.172** |
| - | Res2Net-26w8s [90] | ✓ | 9.3 | 1.45 | 0.147 | 2.72 | 0.272 | 1.47 | 0.169 | - | - |
| - | H/ASP(AAM-softmax) [31] | ✓ | 8.0 | 1.15 | - | 2.49 | - | 1.35 | - | - | - |
| - | H/ASP(AP+softmax) [31] | ✓ | 8.0 | 0.88 | - | 2.21 | - | **1.07** | - | - | - |
| - | ECAPA-TDNN [15] | ✓ | 6.2 | 1.01 | 0.127 | 2.32 | 0.218 | 1.24 | 0.142 | - | - |
| - | MFA-TDNN (standard) [41] | ✓ | 7.32 | **0.856** | 0.092 | 2.049 | 0.190 | 1.083 | 0.118 | - | - |
| - | MFA-TDNN (lite) [41] | ✓ | 5.93 | 0.968 | 0.091 | 2.174 | 0.199 | 1.138 | 0.121 | | |
| - | ECAPA-TDNN(MBFA-MW) [61] | ✓ | - | 0.87 | 0.115 | 2.31 | 0.222 | 1.22 | 0.135 | - | - |
| - | UP-LS UP-PLDA [79] | ✓ | - | 1.01 | 0.124 | 2.18 | 0.224 | - | - | 1.59 | 0.167 |
| 6 | ECAPA-TDNN [15]† | ✓ | 6.19 | 1.127 | 0.145 | 2.249 | 0.223 | 1.194 | 0.133 | 1.668 | 0.155 |
| 7 | ResNet34 [77]† | ✓ | 6.63 | 1.101 | 0.128 | 2.221 | 0.208 | 1.252 | 0.139 | 1.584 | 0.161 |
| 8 | Xi [33] + **proposed tResNet**‡ | ✓ | 6.54 | 0.936 | 0.115 | 1.942 | 0.186 | 1.110 | 0.125 | 1.394 | 0.145 |
| 9 | **RecXi**($\tilde{\phi}, \tilde{\phi}_{\text{lin}}$) with $\mathcal{L}_{\text{ssp}}$ + tResNet | ✓ | 7.06 | 0.984 | **0.091** | **1.857** | **0.179** | 1.075 | **0.114** | **1.340** | **0.137** |

## 5  Results and Discussions

In Table 1, Table 2 and 3, we report the performance in terms of EER and minDCF of the proposed RecXi, various baselines and SOTA systems. It is worth noting that the VoxCeleb1-O test set is much smaller and easier compared to the other three test sets, which may lead to less convincing results.

**Comparison with SOTA methods and baselines.** In Table 1, we compare the baseline systems and proposed RecXi systems with SOTA methods. The baseline systems with proposed the tResNet backbone (system #3 and #4) achieve the SOTA performance. Compared to the TSP method (system #3), the use of Xi (system #4) achieves overall improvements with the 5.02%/4.19% average reductions in EER/minDCF. This also renders further performance improvements more difficult. By comparing systems #4 and #5, we observe that the proposed RecXi leads to relative improvements with 6.96%/5.82% average reductions in EER/minDCF compared to the Xi baseline system. The proposed RecXi (system #5) obviously further improves the performance.

**Experiments with data augmentation.** In the second half of Table 1, we report the results with augmented data (see Appendix C.2). The re-implemented ECAPA-TDNN (system #6) achieves slightly better performance than that reported in [15]. Consistent with the conclusions of training without data augmentation discussed above, the baseline (system #8) with proposed tResNet outperforms all SOTA methods and the proposed RecXi (system #9) shows great superiority over all other systems.

**Comparison for ECAPA-TDNN backbone-based systems.** The experiments discussed above are for Conv2D-based networks. For speech-related tasks, the TDNN-based networks are widely used as well [74, 15, 41, 53]. In order to verify the superiority of the proposed method under different conditions, the experiment results for the systems based on the popular ECAPA-TDNN [15] backbone

Table 2: Performance in EER(%) and minDCF of the baselines and proposed RecXi with **ECAPA-TDNN backbone** [15] on VoxCeleb1 and SITW test sets. # is the index number for the system.

| # | Aggregation Layer | Params (Million) | VoxCeleb1-O EER | minDCF | VoxCeleb1-H EER | minDCF | VoxCeleb1-E EER | minDCF | SITW *eval* EER | minDCF |
|---|---|---|---|---|---|---|---|---|---|---|
| 1 | Chan.&Con. | 6.19 | 1.377 | 0.137 | 2.674 | 0.235 | 1.451 | 0.153 | 2.542 | 0.208 |
| 10 | Xi | 5.90 | 1.340 | 0.140 | 2.647 | 0.237 | 1.429 | 0.148 | 2.679 | 0.204 |
| 11 | **RecXi($\tilde{\phi}, \tilde{\phi}_{\text{lin}}$) with $\mathcal{L}_{\text{ssp}}$** | 6.43 | **1.196** | **0.107** | **2.467** | **0.227** | **1.292** | **0.141** | **2.105** | **0.184** |

are reported in Table 2. Compared to the original aggregation layer of ECAPA-TDNN (system #1), the use of xi-vector posterior inference (system #10) performs slightly better. The comparison between system #10 and #11 presents that for the ECAPA-TDNN backbone, the proposed RecXi achieves average EER/minDCF reductions of 12.15%/10.66% over the Xi baseline.

**Brief summary.** Since the Xi baseline performs the best among all baselines and SOTA systems discussed above with different training conditions and backbone networks, we refer to it as a SOTA baseline for the following discussion. We observe that compared to the SOTA baseline, the proposed RecXi consistently improves the performance for both backbone networks on all four test sets with overall average EER/minDCF reductions of 9.56%/8.24% (system #4 vs. #5 and #10 vs. #11). The experiment results remain consistent regardless of the backbone types and whether augmented data is used. This may be due to the fact that the proposed disentanglement framework disentangles the static speaker components effectively and benefits speaker recognition. As the speaker is disentangled under the assistance of disentangled content representation, the significant improvement also proves the quality of the dynamic counterpart modeling. In addition, we found that the performance of tResNet-based systems is generally better than that of the systems with ECAPA-TDNN backbone.

Table 3: Performance in EER(%) and minDCF of various RecXi systems on VoxCeleb1 and SITW test sets for ablation study. # is the index number for the system. BN and Para. indicate the type of backbone network and number of parameters in million, respectively.

| # | BN | Aggregation Layer | $\mathcal{L}_{\text{ssp}}$ | Para. | VoxCeleb1-H EER | minDCF | VoxCeleb1-E EER | minDCF | SITW *eval* EER | minDCF | Relative Reduction EER | minDCF |
|---|---|---|---|---|---|---|---|---|---|---|---|---|
| 11 | ECAPA | RecXi($\tilde{\phi}, \tilde{\phi}_{\text{lin}}$) | ✓ | 6.43 | 2.467 | **0.227** | 1.292 | 0.141 | **2.105** | **0.184** | -7.58% | -12.18% |
| 12 | ECAPA | RecXi($\tilde{\phi}$) | ✓ | 6.13 | **2.445** | 0.233 | **1.286** | **0.139** | 2.160 | 0.191 | -7.30% | -11.04% |
| 13 | ECAPA | RecXi($\tilde{\phi}, \tilde{\phi}_{\text{lin}}$) | ✗ | 6.43 | 2.477 | 0.228 | 1.326 | 0.142 | 2.351 | 0.196 | -3.41% | -10.40% |
| 14 | ECAPA | RecXi($\tilde{\phi}$) | ✗ | 6.13 | 2.498 | 0.229 | 1.325 | 0.146 | 2.597 | 0.273 | Benchmark | |
| 5 | tResNet | RecXi($\tilde{\phi}, \tilde{\phi}_{\text{lin}}$) | ✓ | 7.06 | **2.097** | 0.196 | **1.197** | **0.124** | 1.832 | **0.172** | -5.78% | -5.83% |
| 15 | tResNet | RecXi($\tilde{\phi}$) | ✓ | 6.73 | 2.117 | **0.192** | 1.215 | 0.128 | **1.750** | 0.177 | -6.35% | -4.82% |
| 16 | tResNet | RecXi($\tilde{\phi}, \tilde{\phi}_{\text{lin}}$) | ✗ | 7.06 | 2.130 | 0.198 | 1.222 | 0.128 | 1.886 | 0.184 | -3.71% | -2.38% |
| 17 | tResNet | RecXi($\tilde{\phi}$) | ✗ | 6.73 | 2.185 | 0.204 | 1.230 | 0.128 | 2.050 | 0.193 | Benchmark | |

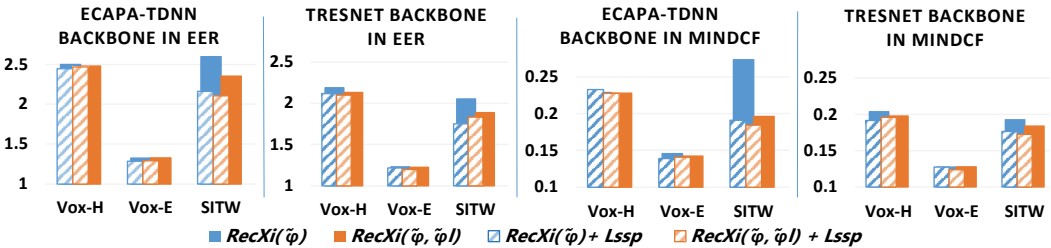

Figure 2: Bar charts for ablation studies. Performance in EER(%) and minDCF of proposed RecXi under different conditions are drawn. The blue and orange each indicate RecXi($\tilde{\phi}$) and RecXi($\tilde{\phi}, \tilde{\phi}_{\text{lin}}$). Patterned color bars at the left front and solid color bars at the right behind represent the performance with and without $\mathcal{L}_{\text{ssp}}$ for the same test trial, respectively. For all bars, the shorter the better.

**Ablation Study.** We perform ablation studies on the proposed RecXi. As the results shown in Table 3 are detailed but not intuitive, we show Figure 2 for a better view. From the charts, the following conclusions are summarized and are consistent for both ECAPA-TDNN and tResNet backbones:

1) As illustrated in Figure 2, by comparing the solid color bars with the patterned color bars while ignoring the difference of the colors, we find that most patterned color bars are shorter. It indicates that the systems with proposed self-supervision $\mathcal{L}_{\mathrm{ssp}}$ perform better than those without. It also proves the effectiveness of $\mathcal{L}_{\mathrm{ssp}}$ for RecXi frameworks. The results in Table 3 shows that the proposed self-supervision $\mathcal{L}_{\mathrm{ssp}}$ leads to an overall 5.07%/5.44% average reductions (system #11 vs. #13, #12 vs. #14, #5 vs. #16, #15 vs. #17) in EER/minDCF for all RecXi systems.

2) By comparing RecXi($\tilde{\phi}$) shown as the solid blue color bars and RecXi ($\tilde{\phi}$, $\tilde{\phi}_{\mathrm{lin}}$) shown as solid orange color bars, we conclude that when $\mathcal{L}_{\mathrm{ssp}}$ is not applied, the systems derived only from $\tilde{\phi}$ perform worse than those using both $\tilde{\phi}$ and $\tilde{\phi}_{\mathrm{lin}}$ in most of the trials (system #13 vs. #14 and #16 vs. #17). When we consider the patterned color bars, however, we find that these gaps are overcome, and very similar performance is achieved when $\mathcal{L}_{\mathrm{ssp}}$ is applied. We believe that the gaps between RecXi($\tilde{\phi}$)-based systems and RecXi($\tilde{\phi}$, $\tilde{\phi}_{\mathrm{lin}}$)-based systems are caused by the constraints on the content disentanglement layer provided by $\tilde{\phi}_{\mathrm{lin}}$. This brings RecXi ($\tilde{\phi}$, $\tilde{\phi}_{\mathrm{lin}}$) systems an advantage in content disentanglement and further improves the efficiency of disentangling speaker representation. When both kinds of systems are enhanced by the self-supervision loss $\mathcal{L}_{\mathrm{ssp}}$, a sufficient guide for disentangling content representations is provided, and it leads to a similar and improved performance. It shows the effectiveness and superiority of the proposed $\mathcal{L}_{\mathrm{ssp}}$ and also proves that the speaker embedding benefits from the quality of disentangled content.

Additional experiments are provided in the Appendix, covering diverse aspects including visualization, the ablation study on three RecXi layers, comparisons between RecXi and SOTA systems using ASR models or contrastive learning, as well as the evaluation of the effectiveness of $\mathbf{G}_t$. For details, please refer to Appendix F, G, H, and I, respectively.

## 6 Limitations

This work has some limitations: 1). The novel $\mathcal{L}_{\mathrm{ssp}}$ is intuitive and effective but needs further investigation and improvement. 2). As mentioned in Appendix C.2, the number of mini-transition models for deriving $\mathbf{G}_t$ is set as $N = 16$. In Appendix I, we verify the effectiveness and necessity of utilizing the proposed $\mathbf{G}_t$. However, a comprehensive investigation of this hyperparameter is not conducted. This hyperparameter can be further exploited as it is related to the acoustic features and dynamic components we wish to disentangle.

The discussion about broader impacts is available in Appendix E.

## 7 Conclusion and Future Work

We propose the RecXi – a Gaussian inference-based disentanglement learning neural network. It models the dynamic and static components in speech signals with the aim to disentangle vocal and verbal information in the absence of text labels and benefit the speaker recognition task. In addition, a novel self-supervised speaker-preserving method is proposed to relieve the effect of text labels absent for fine content representation disentanglement. The experiments conducted on both VoxCeleb and SITW datasets prove the consistent superiority of the proposed RecXi and the effectiveness of the proposed self-supervision method. We expect that the proposed model is applicable to automatic speech recognition (ASR), where speaker-independent representation is desirable. In addition, the disentangled content and speaker embeddings are useful in the voice conversion and speech synthesis tasks. For future work, we plan to reconstruct speech signals and utilize the interaction between these two tasks to benefit each other.

## 8 Acknowledgements

This work is supported by the Agency for Science, Technology and Research (A⋆STAR), Singapore, through its Council Research Fund (Project No. CR-2021-005). Haizhou Li is in part supported by the National Natural Science Foundation of China (Grant No. 62271432) and the Agency for Science, Technology and Research (A⋆STAR) under its AME Programmatic Funding Scheme (Project No. A18A2b0046). The authors would like to thank the reviewers and the meta-reviewer for their comments, which greatly improved the article.

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
