# Disentangling Voice and Content with Self-Supervision for Speaker Recognition
# (Appendix)

**Tianchi Liu[1,2], Kong Aik Lee[3,1], Qiongqiong Wang[1], Haizhou Li[4,2]**

[1] Institute for Infocomm Research (I²R), Agency for Science, Technology and Research (A⋆STAR), Singapore
[2] Dept. of Electrical and Computer Engineering, National University of Singapore, Singapore
[3] Dept. of Electrical and Electronic Engineering, Hong Kong Polytechnic University, Hong Kong
[4] School of Data Science, The Chinese University of Hong Kong, Shenzhen, China
{liu_tianchi, wang_qiongqiong}@i2r.a-star.edu.sg, kongaik.lee@ieee.org, haizhouli@cuhk.edu.cn

## A Simplified Implementation for Gaussian Inference in RecXi

In this section, we will introduce the simplified method for implementing the proposed Gaussian inference. We take *layer 2* as an example and Equations (19) and (20) are repeated as follows:

$$\rho_t = \mathbf{\Phi}_t^{-1}[(\mathbf{L}_t^{-1} + (\mathbf{P}_t^+)^{-1})^{-1}(\mathbf{z}_t - \phi_t^+) + \mathbf{\Phi}_{t-1}^+\rho_{t-1}], \tag{Appx.1}$$

$$\mathbf{\Phi}_t = (\mathbf{L}_t^{-1} + (\mathbf{P}_t^+)^{-1})^{-1} + \mathbf{\Phi}_{t-1}^+. \tag{Appx.2}$$

We formulate the adjusted uncertainty estimate $\mathbf{L}'_t$ and point estimation with speaker representation removal $\mathbf{z}'_t$ as following:

$$\mathbf{L}'_t = (\mathbf{L}_t^{-1} + (\mathbf{P}_t^+)^{-1})^{-1}, \tag{Appx.3}$$

$$\mathbf{z}'_t = \mathbf{z}_t - \phi_t^+. \tag{Appx.4}$$

Then, Equations (Appx.1) and (Appx.2) are simplified as

$$\rho_t = \mathbf{\Phi}_t^{-1}\mathbf{L}'_t\mathbf{z}'_t + \mathbf{\Phi}_t^{-1}\mathbf{\Phi}_{t-1}^+\rho_{t-1}, \tag{Appx.5}$$

$$\mathbf{\Phi}_t = \mathbf{L}'_t + \mathbf{\Phi}_{t-1}^+. \tag{Appx.6}$$

Similar to [9], we assume that the covariance (and precision) matrices are diagonal and choose to estimate directly the log-precision which turns out to be more convenient for following derivation. And $\mathbf{L}'_t$ in (Appx.3) can be simplified by computing the diagonal elements directly and thereby avoiding the expensive computational for matrix inverse operations. The i-th diagonal element is formulated as

$$\mathbf{L}'_{t(i)} = \frac{\mathbf{L}_{t(i)}\mathbf{P}_{t(i)}}{\mathbf{L}_{t(i)} + \mathbf{P}_{t(i)}}. \tag{Appx.7}$$

Let gain factor $\mathbf{A}_t = \mathbf{\Phi}_t^{-1}\mathbf{L}'_t$ and $\mathbf{A}_{t-1} = \mathbf{\Phi}_t^{-1}\mathbf{\Phi}_{t-1}^+$, then,

$$\mathbf{A}_t = \mathbf{\Phi}_t^{-1}\mathbf{L}'_t = [\mathbf{L}'_t + \mathbf{\Phi}_{t-1}^+]^{-1}\mathbf{L}'_t, \tag{Appx.8}$$

$$\mathbf{A}_{t-1} = \mathbf{\Phi}_t^{-1}\mathbf{\Phi}_{t-1}^+ = [\mathbf{L}'_t + \mathbf{\Phi}_{t-1}^+]^{-1}\mathbf{\Phi}_{t-1}^+. \tag{Appx.9}$$

Therefore, the i-th diagonal element of the gain factor is computed as

$$\mathbf{A}_{t(i)} = \frac{\mathbf{L}'_{t(i)}}{\mathbf{L}'_{t(i)} + \mathbf{\Phi}_{t-1(i)}^+} = \frac{\exp(\log(\mathbf{L}'_{t(i)}))}{\exp(\log(\mathbf{L}'_{t(i)})) + \exp(\log(\mathbf{\Phi}_{t-1(i)}^+))}, \tag{Appx.10}$$

$$\mathbf{A}_{t-1(i)} = \frac{\mathbf{\Phi}_{t-1(i)}^+}{\mathbf{L}'_{t(i)} + \mathbf{\Phi}_{t-1(i)}^+} = \frac{\exp(\log(\mathbf{\Phi}_{t-1(i)}^+))}{\exp(\log(\mathbf{L}'_{t(i)})) + \exp(\log(\mathbf{\Phi}_{t-1(i)}^+))}. \tag{Appx.11}$$

37th Conference on Neural Information Processing Systems (NeurIPS 2023).

Let $\mathbf{K} = [\log(\mathbf{L}'_{t(i)}), \log(\mathbf{\Phi}^+_{t-1(i)})]$, then $\mathbf{A}_{t(i)}$ and $\mathbf{A}_{t-1(i)}$ equal to the outputs of a *softmax* function ($\sigma$) of $\mathbf{K}$.

$$\mathbf{A}_{t(i)} = \sigma(\mathbf{K})_1, \qquad (Appx.12)$$

$$\mathbf{A}_{t-1(i)} = \sigma(\mathbf{K})_2. \qquad (Appx.13)$$

As mentioned above, the covariance and precision matrices are estimated directly the log-precision, therefore, $\log(\mathbf{L}'_t)$ and $\log(\mathbf{\Phi}^+_{t-1})$ are estimated and used directly in (Appx.12) and (Appx.13). And Equation (Appx.5) can be simplified as

$$\rho_t = \mathbf{A}_t \mathbf{z}'_t + \mathbf{A}_{t-1}\rho^+_{t-1}. \qquad (Appx.14)$$

As the gain factor $\mathbf{A}$ is a diagonal matrix, and $\mathbf{z}$ and $\phi$ are vectors, the expensive matrix multiplication operations and numerically problematic matrix inversions are simplified into element-wise multiplication of diagonal elements and vectors. This is the same as the implementation of point-wise multiplication for matrices in neural networks and thus, is easy to implement based on existing toolkits.

The method above can also be applied to *layer 1* and *layer 3* of the proposed RecXi. We can simplify the implementation of the entire RecXi framework by avoiding the complex and computationally expensive matrix inverse and matrix multiplication operations.

## B  Comparison between the Modified ResNet34 and Proposed tResNet34

As mentioned in Section 4.2, to model more local regions with larger frequency bandwidths at different scales, we further modify the ResNet34 backbone used in [20] by simply changing the stride strategy and naming it tResNet. The structure and outputs for both backbone models are detailed in Table Appx.1.

Table Appx.1: The structure and outputs comparison between modified ResNet34 in [20] and proposed tResNet34. The main difference is the stride strategy.

| Layer | | ResNet34 | | tResNet34 | |
|---|---|---|---|---|---|
| | | stride | Output Size | stride | Output Size |
| $3 \times 3, 32$ | | (1,1) | $32 \times F \times T$ | (1,1) | $32 \times F \times T$ |
| $\begin{bmatrix} 3 \times 3, 32 \\ 3 \times 3, 32 \end{bmatrix} \times 3$ | | (1,1) | $32 \times F \times T$ | (2,1) | $32 \times F/2 \times T$ |
| $\begin{bmatrix} 3 \times 3, 64 \\ 3 \times 3, 64 \end{bmatrix} \times 4$ | | (2,2) | $64 \times F/2 \times T/2$ | (2,1) | $64 \times F/4 \times T$ |
| $\begin{bmatrix} 3 \times 3, 128 \\ 3 \times 3, 128 \end{bmatrix} \times 6$ | | (2,2) | $128 \times F/4 \times T/4$ | (2,2) | $128 \times F/8 \times T/2$ |
| $\begin{bmatrix} 3 \times 3, 256 \\ 3 \times 3, 256 \end{bmatrix} \times 3$ | | (2,2) | $256 \times F/8 \times T/8$ | (2,1) | $256 \times F/16 \times T/2$ |

The testing results of two backbone models are stated in Table Appx.2. Temporal statistics pooling is applied for both backbones, which is also the default aggregation layer used in [20].

Table Appx.2: Performance in EER(%) and minDCF of the ResNet34 in [20] and proposed tResNet34 on VoxCeleb1 and SITW test sets. # is the index number for the system.

| # | Backbone | params (Million) | VoxCeleb1-O | | VoxCeleb1-H | | VoxCeleb1-E | | SITW *eval* | |
|---|---|---|---|---|---|---|---|---|---|---|
| | | | EER | minDCF | EER | minDCF | EER | minDCF | EER | minDCF |
| 18 | ResNet34 | 6.63 | 1.489 | 0.155 | 2.500 | 0.224 | 1.423 | 0.158 | 2.378 | 0.208 |
| 3 | tResNet34 | **6.21** | **1.396** | **0.135** | **2.257** | **0.204** | **1.281** | **0.141** | **2.250** | **0.200** |

The tResNet design is not claimed as one of the major contributions of this work. A thorough investigation and exploration of the tResNet mechanism will be presented in a forthcoming work.

# C Experiments Setup

## C.1 Dataset

The experiments are conducted on VoxCeleb1 [13], VoxCeleb2 [2], and the Speaker in the Wild (SITW) [12] datasets. During training, only the development partition of the VoxCeleb2 dataset is used. It contains 1,092,009 utterances from 5,994 speakers. VoxCeleb1 and SITW-eval datasets are used as the test sets. There are three test lists in the VoxCeleb1, namely VoxCeleb-O, VoxCeleb-H, and VoxCeleb-E. There is no overlapping speaker between the training and testing sets. In addition, we reserved a small portion of about 2% from the training set as the validation set. It is worth noting that the text labels are not available for these datasets. The number of speakers and utterances for these two datasets are shown in Table Appx.3.

Table Appx.3: Number of speakers and utterances of VoxCeleb1 and VoxCeleb2 dataset.

| Data set | # Speakers | # Utterances |
|----------|------------|--------------|
| VoxCeleb1 | 1,211 | 148,642 |
| VoxCeleb2 | 5,994 | 1,092,009 |

## C.2 Training Strategy

**Implementation** The experiments are conducted using Pytorch[1] and implemented in SpeechBrain[2]. For a fair comparison, the baselines are all re-implemented and trained with the same strategy as RecXi which follows that in ECAPA-TDNN [4] and MFA-TDNN [11]. The details are stated below:

**Configuration** The Adam [6] optimizer with cyclical learning rate scheduler [15] following triangular policy [15] is used for training all models. The maximum and minimum learning rates of the cyclical scheduler are 8e-3 and 8e-8 for ECAPA-TDNN-based systems, 3e-3 and 3e-8 for ResNet-based systems. All the samples are chunked into 3-second segments during training without augmentation. The mini-batch size is 384 for ECAPA-TDNN-based systems and 256 for ResNet-based systems considering the limitations of GPU memory. Each model is trained by two NVIDIA A5000 GPUs or NVIDIA 3090 GPUs with 24GB memory.

A total of 16 epochs are trained for each system and at the end of each epoch, the model is evaluated by the validation set to find the best checkpoint for testing. The criterion of classification loss $\mathcal{L}_{cls}$ is additive angular margin softmax (AAM-softmax) [3] with a margin of 0.2, a scale of 30 and a weight decay of 2e-5. $\alpha$ and $\beta$ in Equation (27) are 1.0 and 3000, respectively, following the default setting in [18].

**Data Augmentation** For the experiments marked with data augmentation, we employ five augmentation techniques to increase the diversity of the training data. The first two follow the idea of random frame dropout in the time domain [14] and speed perturbation [7]. The remaining three are a set of reverberate data, noisy data, and a mixture of both by using RIR dataset [8]. The mini-batch size is 384 (64 original samples each with 5 augmented samples). The maximum and minimum learning rates of the cyclical scheduler are 2e-3 and 2e-8. Each model is trained by 4 or 6 NVIDIA A5000 GPUs each with 24GB memory. The other configurations are the same as the experiments without data augmentation as described above.

**Model details** The number $N$ of mini-transition models for deriving $\mathbf{G}_t$ is set as 16. The bottleneck feature dimension of the filter generator between two fully connected layers is set as 256. The bottleneck feature dimension of uncertainty estimation in the encoder for both xi-vector and RecXi is also 256. The channels in the convolutional frame layers of ECAPA-TDNN backbone is 512 following the default setting in [4].

---

[1]https://pytorch.org/
[2]https://speechbrain.github.io/

For the decoder, we follow the designs in ECAPA-TDNN [4] and ResNet [20] baselines. Specifically, two fully connected (FC) layers following the aggregation layer are employed. For the ECAPA-TDNN-based systems, one batch normalization layer is applied before the first FC layer. The first FC layer produces embeddings and the other one is a classification layer. The embedding dimension is 192 for the ECAPA-TDNN-based systems and 256 for the ResNet/tResNet-based systems. The backbone network is trained together with the aggregation layer RecXi, as well as the decoder.

The code and the pre-trained models will be made available with third-party re-implementation.

### C.3 Evaluation Protocol

All the models are evaluated by the performance in terms of equal error rate (EER) and the minimum detection cost function (minDCF) with $P_{\text{target}} = 0.01$ and $C_{\text{FA}} = C_{\text{Miss}} = 1$. The test trial scores are calculated by measuring the cosine similarity between embeddings. The S-norm [5] post-processing method is applied for all experiments.

## D Comparisons between Different Kinds of $\mathcal{L}_{\text{ssp}}$

Table Appx.4: Performance in EER(%) and minDCF of systems without proposed $\mathcal{L}_{\text{ssp}}$ and with different kinds of $\mathcal{L}_{\text{ssp}}$. The model used is ECAPA-TDNN backbone with proposed RecXi($\tilde{\phi}$, $\tilde{\phi}_{\text{lin}}$). # is the index number for the system.

| # | Loss for Self-supervision | VoxCeleb1-O | | VoxCeleb1-H | | VoxCeleb1-E | | SITW *eval* | |
|---|---|---|---|---|---|---|---|---|---|
| | | EER | minDCF | EER | minDCF | EER | minDCF | EER | minDCF |
| 13 | Not used | 1.303 | 0.116 | 2.477 | 0.228 | 1.326 | 0.142 | 2.351 | 0.196 |
| 19 | Mean Squared Error | 1.202 | 0.128 | 2.471 | 0.228 | 1.296 | **0.136** | 2.292 | 0.184 |
| 11 | Similarity-Preserving | **1.196** | **0.107** | **2.467** | **0.227** | **1.292** | 0.141 | **2.105** | **0.184** |

In this paper, we chose the similarity-preserving (SP) loss [18] as $\mathcal{L}_{\text{ssp}}$, as it is more in line with our layer-wise design and the idea of speaker information preserving. The experiment results reported in Table Appx.4 show that by using either mean squared error (MSE) loss (system #19) or SP loss (system #11) as proposed $\mathcal{L}_{\text{ssp}}$, there is a significant improvement compared to not using $\mathcal{L}_{\text{ssp}}$ (system #13). This proves the effectiveness of our proposed novel self-supervision method both with MSE loss and SP loss. In addition, we can observe that the system with SP loss as $\mathcal{L}_{\text{ssp}}$ outperforms the system with MSE loss as $\mathcal{L}_{\text{ssp}}$ in most of the trials. This shows that SP loss is more appropriate for our design.

Our novelty lies in the approach of using knowledge distillation loss to effectively guide content disentanglement without relying on textual labels, rather than finding the optimal knowledge distillation loss. However, as discussed in Section 6, it is also valuable to explore the appropriate loss function for $\mathcal{L}_{\text{ssp}}$. Therefore, we include Table Appx.4 here for those who may be interested.

## E Broader Impact

This work proposes a Gaussian inference-based disentanglement learning neural network namely RecXi. It models the dynamic and static components in the speech signals with the aim to disentangle vocal and verbal information and benefit speaker recognition. The system is further enhanced by the proposed novel self-supervision $\mathcal{L}_{\text{ssp}}$ method in the absence of text labels. The research shows great effectiveness over a variety of modern methods, and it may have a wide range of other speech-related applications, such as automatic speech recognition (ASR), voice conversion, and anonymization.

On the negative side, even though the proposed method achieves the best performance so far in the field of speaker verification, it is still not perfect and may make wrong decisions at a low possibility. Similar to existing deep learning-based solutions, the decisions made are hard to interpret. This limits its application in some critical applications, such as forensic voice comparison and banking where false acceptances are serious issues. In addition, as mentioned above and in Section 7, our next plan is to reconstruct speech signals from the disentangled representations, which can also be extended to voice conversion. The voice conversion techniques can generate audio with realistic sound, and

there is an increased potential risk of harm, malicious use, and ethical issues [10]. Specifically, these systems could be misused in various manners, such as fake news generation and voice spoofing. To mitigate these issues, audio anti-spoofing (or deepfake audio detection) is studied [17, 21].

# F  Visualization

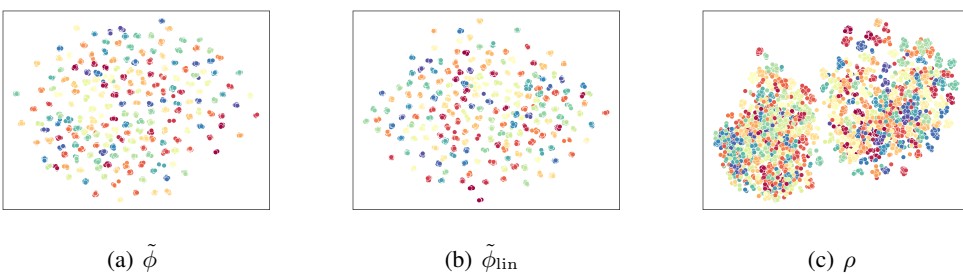

(a) $\tilde{\phi}$          (b) $\tilde{\phi}_{\text{lin}}$          (c) $\rho$

Figure Appx.1: The t-distributed stochastic neighbor embedding (t-SNE) [19] visualization of speaker discriminative ability in embeddings space. (a) The disentangled speaker representation $\tilde{\phi}$ from *layer 3*. (b) The speaker representation $\tilde{\phi}_{\text{lin}}$ by the linear operation in Equation (25). (c) The content representation $\rho$ from *layer 2*.

The visualization in Fig. Appx.1 depicts speakers from the test set of VoxCeleb, where the first 200 speakers (indexed from id10001 to 10200) are included, each with 20 utterances. From the figure, it is evident that in *layer 3*, the disentangled speaker representation $\tilde{\phi}$ is highly discriminative for speakers. Moreover, benefiting from the self-supervision loss, the speaker representation acquired through the linear operation in Equation (25) also exhibits notable speaker discriminative ability, comparable to that derived from *layer 3*. However, for *layer 2* ($\rho$), as its main objective is to disentangle content information, it lacks the discriminative ability observed in *layer 3*.

# G  Ablation Study on Three RecXi Layers

Table Appx.5: Performance in EER(%) and minDCF of using different posteriors from the three layers of RecXi for ablation study. # is the index number for the system.

| # | Posterior | Representation Description | VoxCeleb1-H | | VoxCeleb1-E | | SITW | |
|---|---|---|---|---|---|---|---|---|
| | | | EER | minDCF | EER | minDCF | EER | minDCF |
| 5 | $\tilde{\phi}, \tilde{\phi}_{\text{lin}}$ | Disentangled Speaker & Speaker by Equation (25) | **2.097** | 0.196 | **1.197** | **0.124** | 1.832 | 0.172 |
| 15 | $\tilde{\phi}$ | Disentangled Speaker | 2.117 | **0.192** | 1.215 | 0.128 | **1.750** | 0.177 |
| 20 | $\tilde{\phi}_{\text{lin}}$ | Speaker by a linear operation (Equation (25)) | 2.181 | 0.198 | 1.222 | 0.126 | 1.804 | **0.172** |
| 21 | $\rho$ | Disentangled Content | 49.421 | 1.000 | 49.022 | 1.000 | 49.399 | 1.000 |
| 22 | $\phi$ | Precursor Speaker | 2.187 | 0.199 | 1.249 | 0.131 | 2.023 | 0.186 |

In order to explore the information within each of RecXi's three layers and the speaker representation achieved through a linear operation in Equation (25), we generate embeddings using the output posteriors of these layers and evaluate their speaker discriminative ability. The results in Table Appx.5 clearly show that both $\tilde{\phi}$ and $\tilde{\phi}_{\text{lin}}$ (systems #5, #15, and #20) carry speaker information and demonstrate great discriminative capabilities. Additionally, reporting on *layer 2* (system #21) further supports our claims, confirming its role in representing content while effectively removing speaker-related information. The EER being close to the maximum value, 50%, indicates that *layer 2* does not contain any speaker-related information and does not exhibit any speaker discriminative ability. The precursor speaker representation $\phi$ (system #22) also exhibits fine speaker discriminative ability, but it is slightly inferior to $\tilde{\phi}$ (system #15). These observations align with those in Fig. Appx.1 and strongly support our claims. Authors express their deep gratitude to anonymous reviewers for inspiring them to conduct this ablation study, which unmistakably demonstrates the success of our disentanglement approach.

## H Comparing RecXi with ASR- or Contrastive Learning-based Systems

Table Appx.6: Performance in EER(%) and minDCF of the proposed RecXi and the SOTA systems with pre-trained ASR models [22, 1] or contrastive learning [16]. # is the index number for the system.

| # | System | Params (Million) | Need Speaker Labels? | Need Pre-trained ASR Model? | VoxCeleb1-O EER | minDCF | VoxCeleb1-H EER | minDCF | VoxCeleb1-E EER | minDCF |
|---|--------|-----------------|---------------------|-----------------------------|-----------------|--------|-----------------|--------|-----------------|--------|
| 23 | IPA [22] | >60 | ✓ | ✓ | 1.81 | - | 3.12 | - | 1.68 | - |
| 24 | NEMO [1] | 15.88 | ✓ | ✗ | 0.88 | 0.137 | 2.20 | 0.225 | 1.08 | 0.134 |
| 25 | NEMO [1] | 15.88 | ✓ | ✓ | **0.74** | 0.110 | 1.90 | 0.189 | **0.90** | **0.105** |
| 26 | MCL-DPP [16] | 10.5 | ✗ | ✗ | 2.89 | - | 6.27 | - | 3.17 | - |
| 27 | MCL-DPP-C [16] | 10.5 | Pseudo label | ✗ | 1.44 | - | 3.27 | - | 1.77 | - |
| 9 | **Proposed RecXi** | **7.06** | ✓ | ✗ | 0.984 | **0.091** | **1.857** | **0.179** | 1.075 | 0.114 |

As introduced in Section 1, pre-trained ASR models have been shown to be beneficial for the speaker recognition task. In this appendix section, we perform a comparison between our proposed method and the SOTA systems in two aspects: 1) using ASR models to provide phonetic information for speaker recognition [22], and 2) utilizing ASR models as initial weights [1]. It's worth noting that [1] is a recent work that became accessible after our submission. This comparison is added during the rebuttal phase.

As shown in Table Appx.6, our proposed system #9 not only surpasses the system employing a pre-trained ASR model (system #23) but also significantly reduces the model size. Additionally, the proposed method achieves similar performance with the model utilizing ASR pre-training in [1] (system #9 vs #25). Our proposed method offers a significant advantage: it achieves competitive performance without requiring pre-training of an ASR model. Additionally, our method is approximately 55.5% smaller than that in [1], highlighting the efficiency and effectiveness of the proposed RecXi.

Contrastive learning has been extensively investigated and has demonstrated great performance in speaker verification. It holds the advantage of effectively utilizing unlabeled data [16]. We also compare the proposed RecXi and a SOTA contrastive learning system, both evaluated on the same dataset. Notably, even though the models in [16] incorporate extra visual information beyond speech, our proposed RecXi consistently demonstrates substantial superiority over the system detailed in [16] across all three test sets.

## I Evaluation of the Effectiveness of $\mathbf{G}_t$

Table Appx.7: Performance in EER(%) and minDCF of using different numbers ($N$) of learnable transition matrices for frame-wise content-aware transition model $\mathbf{G}_t$ or identity matrix follows that of the xi-vector. This aims to verify the effectiveness of $\mathbf{G}_t$, rather than tuning the hyperparameter $N$. # is the index number for the system.

| # | $\mathcal{L}_{\mathrm{ssp}}$ | Posterior | Number of Learnable Transition Matrices ($N$) | VoxCeleb1-H EER | minDCF | VoxCeleb1-E EER | minDCF | SITW EER | minDCF |
|---|------|-----------|-----------------------------------------------|-----------------|--------|-----------------|--------|----------|--------|
| 5 | ✓ | $\tilde{\phi}, \tilde{\phi}_{\mathrm{lin}}$ | 16 | **2.097** | **0.196** | **1.197** | **0.124** | **1.832** | **0.172** |
| 28 | ✓ | $\tilde{\phi}, \tilde{\phi}_{\mathrm{lin}}$ | 6 | 2.099 | 0.198 | 1.215 | 0.128 | 1.968 | 0.184 |
| 29 | ✓ | $\tilde{\phi}, \tilde{\phi}_{\mathrm{lin}}$ | 1 | 2.489 | 0.238 | 1.356 | 0.149 | 3.882 | 0.435 |
| 30 | ✓ | $\tilde{\phi}, \tilde{\phi}_{\mathrm{lin}}$ | 0 (use identity matrix) | 2.524 | 0.260 | 1.471 | 0.165 | 4.280 | 0.435 |

To verify the effectiveness of $\mathbf{G}_t$, we conduct experiments by replacing the $\mathbf{G}_t$ with a single learnable matrix (system #29), or an identity matrix (system #30) following that used in the xi-vector, while maintaining the three-layer design with the proposed self-supervision loss. Based on the results shown in Table Appx.7, it is evident that using an identity matrix or a single learnable matrix leads to poor performance. This clearly demonstrates that if the second layer lacks the capacity to model dynamic patterns, the model becomes confused. This, in turn, has a significant impact on *layer 3*'s ability to obtain accurate speaker embeddings, as it relies on removing the content information provided by *layer 2*. The aforementioned observations further validate the effectiveness of the proposed $\mathbf{G}_t$.

Furthermore, when $N$ is set to 6 (system #28), the model achieves performance that is better than the xi-vector baseline (system #4 in Table 1) and is close to the system with $N = 16$ (system #5) but slightly worse. This experiment demonstrates the effectiveness and necessity of our proposed $\mathbf{G}_t$ in facilitating the capability of *layer 2* to model dynamic components.