# OpenReview forum: "Disentangling Voice and Content with Self-Supervision for Speaker Recognition"
_NeurIPS.cc/2023/Conference — NeurIPS 2023 poster_

### Official Review · Reviewer_P6Zh · 2023-07-03

**Soundness:** 3 good
**Presentation:** 2 fair
**Contribution:** 3 good
**Rating:** 7
**Confidence:** 1

**Summary:**

In this paper, the authors proposed RecXi, a framework that disentangles the voice and content for speaker recognition. Also, the authors used self-supervised learning methods to improve performance. The authors focused on the fact that speaker information is static and the content information is dynamic, so the main idea is the method to decompose static information and dynamic information. To achieve their goal, they designed a three-layer Gaussian inference framework. They argued that more accurate speaker representation is decomposed from layer 1 to layer 3 because they extract the content representation in layer 2.

In the experiment section, the authors compared the proposed model with previous works and show the proposed model outperforms several different datasets.

**Strengths:**

- Disentangling different features clearly with self-supervised learning is a very challenging task in many domains. The authors sapientially solve this problem by aiming at the temporal characteristics of the speaker and content.

- The proposed method outperforms the existing works.

**Weaknesses:**

- The authors argued that more accurate speaker representation is decomposed from layer 1 to layer 3. However, it is not verified with the ablation study.

**Questions:**

- It is a novel method but it is quite complex to understand intuitionally. Could you provide more visualized results or example audio for the supplementary material?

**Limitations:**

- In this architecture, it is very difficult to interpret how the model decomposes different features.

---

> ### Author Rebuttal · Authors · 2023-08-08
>
> Dear Reviewer,
> Thank you for taking the time to review our work and providing us with your valuable suggestions and generous comments! **More experiments are presented in the PDF file in the ‘Global’ rebuttal following this year’s NeurIPS guideline.** Please kindly let us know during discussion if you can’t find or access that PDF file. And following is our rebuttal to address your concerns.
>
>
> **Weaknesses**: Thanks for this insightful comment. We have included the ablation study in Table 2 in the PDF. The results clearly show that the speaker representation decomposed in layer 3 provides better discriminative ability than that in layer 1.
>
> Additionally, inspired by this, we further tested the content representation from layer 2. The results further support our claims, confirming its role in representing content while effectively removing speaker-related information.  The EER being close to the maximum value, 50%, indicates that layer 2 does not contain any speaker-related information and does not exhibit any speaker discriminative ability.
>
> We express our deep gratitude to the reviewer for inspiring us to conduct this ablation study, which unmistakably demonstrates the success of our disentanglement approach.
>
>
> **Questions & Limitations**:  Thanks for this great comment. This work focuses on disentangling speaker voice and content information from speech, which proves beneficial for the speaker verification task. As mentioned in the Conclusion and Broader Impact, the disentangled speaker and content representations hold potential for reconstructing the speech signal and enabling voice conversion. And we intend to explore these possibilities in future research.
>
> Following your kind suggestion, we provide the visualization using t-SNE in Figure 1 in the PDF. The speakers are taken from the test set of VoxCeleb. The visualization depicts speakers from the test set of VoxCeleb, where the first 200 speakers (indexed from id10001 to 10200) are included, each with 20 utterances. From the figure, it is evident that in layer 3, the disentangled speaker representations ($\tilde{\phi}$) are highly discriminative for speakers. Moreover, benefiting from the self-supervision loss, the speaker representation $\tilde{\phi_{l}}$ acquired through the linear operation (Eq. 24) also exhibits notable speaker discriminative ability, comparable to that derived from layer 3 ($\tilde{\phi}$). However, for layer 2 ($\rho$), as its main objective is to disentangle content information, it lacks the discriminative ability observed in layer 3 ($\tilde{\phi}$). These observations align with those in Table 2 of the attached PDF and strongly support our claims. We appreciate this great suggestion.
>
> We sincerely express our gratitude for your valuable suggestions, which have been a great source of inspiration and have significantly contributed to the revision of this work.

---

> > ### Comment · Reviewer_P6Zh · 2023-08-21
> >
> > Thank you to the reviewers for providing proper rebuttals. If accepted, I hope the authors include the additional insights they shared through this rebuttal in the final draft.

---

> > > ### Author Response · Authors · 2023-08-21
> > >
> > > Dear reviewer,
> > >
> > > We appreciate your response and the time you've taken to review our rebuttal. Your thoughtful suggestions on visualizations and ablation studies across the three layers have motivated us to conduct additional experiments to gather more evidence supporting our claims. Your input significantly contributes to enhancing the paper's quality. If this paper is accepted, we will include the rebuttal content in the final manuscript.
> > >
> > > Warm regards,
> > >
> > > Authors

---

### Official Review · Reviewer_krxU · 2023-07-06

**Soundness:** 3 good
**Presentation:** 3 good
**Contribution:** 3 good
**Rating:** 6
**Confidence:** 4

**Summary:**

This paper describes a novel speaker representation based on a three layer Gaussian inference network for speaker recognition.  The model approach uses a static representation for layers 1 and 3, but a time-variant representation for layer 2.  The authors position this network structure as disentangling speaker and content information, as speaker information is constant across a speech signal, while content information is time-varying.

**Strengths:**

The motivation is well justified.  The description of the approach is well written and relatively easy to follow.

Good details are included in Table 1 about which models are reimplemented and which are reported from published results.

The paper includes a nice ablation to understand the contribution of l_ssp and the availability of both \phi and \phi_t


**Weaknesses:**

The best performing model includes \phi and \phi_t both the static and dynamic components for speaker representation.  This has two limitations – first, it makes for a much larger representation vector, and the length of the vector is dependent on T (the max length) being fixed across samples, second, it somewhat undermines the central claim of the proposed technique that the static \phi should contain speaker information while the dynamic \phi_t should isolate content contributions.

The introduction claims that using pretrained ASR to isolate content is suboptimal because it requires too many parameters.  Therefore the aim of this work is to isolate content from the representation with fewer parameters than a full ASR model and without text labels available during training.  This is a fair set of constraints, but it would be useful to have a comparison to a model that uses an explicit content-aware decomposition.  That is, how well does a speaker recognition model like those enumerated in the Introduction that uses an ASR model, even at the cost of additional computation, perform at this task?


**Questions:**

Would a 5 layer network provide improved quality by including a second layer of time-varying inference as layer 4 and a third static layer as layer 5 to further refine the static/time variant representations? (if so, what about 7? etc.

Is T, the maximum length fixed a priori for all samples? If so, are RecXi representations robust to different utterance lengths with varying T?

How is N tuned in the Frame-wise Content-aware Transition model G_t?

Page 5 Section 3.2 re Layer 3: how expensive are the matrix multiplications such that simplification is necessary here? How much improvement to computation (FLOPS? latency?) are these optimizations providing?

Table 1: Are 4 significant digits necessary to understand the results? many include only 2.  It might be clearer to be consistent across all results.

Any concern about other channel characteristics, noise, or reverberant environment characteristics being captured by the static representation?


**Limitations:**

Limitations are included in the last Appendix in the supplemental material.   Per https://neurips.cc/public/guides/PaperChecklist "You are encouraged to create a separate "Limitations" section in your paper."  I'd encourage the authors to include this section in the body of the paper, but the limitations are raised nonetheless

---

> ### Author Rebuttal · Authors · 2023-08-08
>
> Dear reviewer:
>
> We sincerely appreciate your effort in reviewing our work and providing valuable and insightful suggestions. In response to your feedback, **we have conducted additional experiments, and the results are included in the PDF file attached to the 'Global' rebuttal, following this year's NeurIPS guideline.** If you encounter any difficulty accessing the PDF file or have any questions during the discussion, please do not hesitate to inform us. We have carefully considered your concerns, and the following is our rebuttal to address them.
>
> **Weakness 1**:
> The motivation of using $\tilde{\phi_{l}}$ is to provide supervision when the text label is absent. According to the ablation study presented in Table 3 of the paper, we observe that similar results can be achieved even without $\tilde{\phi_{l}}$ when $l_{ssp}$ is applied, as evident from the comparison between #11 and #12, and #5 and #15.
> Furthermore, we would like to clarify that we use only the last frame of the recursive structure. We will emphasize and clarify this point further in the main body of the paper. Thank you for bringing this to our attention, and we apologize for any misunderstanding.
>
> **Weakness 2**:  Thank you for understanding that we didn't compare the ASR model-based systems because of their significantly larger model size compared to the speaker recognition model. Meanwhile, we also agree with the reviewer that including a comparison with the system using the ASR model is more convincing. Therefore, we add the comparison in Table 1 of the uploaded PDF in the 'Global' rebuttal.
>
> This issue of large model size has led to limited adoption of pre-trained ASR models in SOTA methods. We find a work published in 2019 [1] that utilizes pre-trained ASR models, and a comparison is presented in Table 1. Notably, our proposed method (#9) outperforms the approach (#New1) from [1].
>
> In addition, we find recent work (which became available after our submission)  that utilizes a pre-trained ASR model to initialize the speaker verification model [2]. In Table 1, we observe that RecXi outperforms the model without pre-training in [2] (system #9 vs #new2). Additionally, the proposed method achieves similar performance with the model utilizing ASR pre-training in [2] (system #9 vs #new3). Our proposed method offers a significant advantage: it achieves competitive results without requiring the pre-training of an ASR model. Additionally, our method is approximately **55.5% smaller** than the **smallest** model in [2], highlighting its efficiency and effectiveness.
>
> **Question 1**: Thanks for this great comment. We also came up with the same idea when we were designing the RecXi. While the idea of stacking layer 2 and layer 3 seemed promising, we believe that the benefits of additional layer stacking diminish rapidly. The current three-layer configuration has yielded concrete results, effectively supporting our assumptions and claims. We reserve this further investigation as a future work.
>
>
> **Question 2**: The training samples have a fixed length of 3 seconds, as explained in Appendix C.2 Training Strategy (line 709). During testing, the whole sample length is utilized, resulting in varying input lengths (T). RecXi demonstrates robustness to these varying input lengths, as shown in the test results. This training and testing configuration adheres to the standard practice in the majority of ASV work, ensuring a fair comparison.
>
> **Question 3**: Thank you for bringing up this point. In Table 3 of the PDF and in the analysis provided in the global rebuttal, additional investigations related to N have been presented. However, it is important to underscore that **due to the constraints of the rebuttal time, these new experiments were conducted to verify the effectiveness of $\mathbf{G_t}$ rather than fully fine-tuning the hyperparameter N**. As mentioned in Section 6 Conclusion (line 362), N is a hyperparameter that can be further explored as it is  related to the acoustic features and dynamic components we wish to disentangle. We will continue our investigation to find a better way of selecting this hyperparameter based on the textual representation information. However, its value does not impact our core concept, but only affects performance. Further research on this aspect may reveal more insights into dynamic components, making it a valuable subject for future investigation.
>
> **Question 4**: Thank you for your question. Same as the xi-vector, we assume the matrix is diagonal to simplify the computation from a time complexity of O(n^3) to O(n). Meanwhile, the log domain is applied to provide a better numerical stability. The diagonal assumption and log domain utilization are mentioned in Section 3.1.
>
> **Question 5**: Thank you for providing this great suggestion. We will reduce the results in all tables (including the tables in the PDF file) to 3 significant digits for improved clarity and consistency.
>
> **Question 6**: Thanks a lot for this detailed and insightful question.
> The static representation is specifically designed to capture static signal components such as speaker characteristics, channels, and background noise.  However, channels and noise can be effectively handled through data augmentation by the encoder, making the static representation effective for its intended purpose.
>
> **Limitations**: Thank you for bringing this to our attention. In the final draft, we will include a separate section to address the limitations in more detail if the page limit allows.
>
> We extend our heartfelt appreciation for your valuable suggestions, which have been a great source of inspiration and have played a pivotal role in the revision of this work.

---

> > ### Comment · Reviewer_krxU · 2023-08-19
> >
> > Thank you very much for responding to all of my comments and questions.  they've helped me understand the work better

---

> > > ### Author Response · Authors · 2023-08-19
> > >
> > > Dear Reviewer,
> > >
> > > Thank you so much for your kind reply. We're truly glad that our rebuttal has been helpful. Your comments and questions are greatly appreciated and contribute significantly to the quality of our work. Your support means a lot to us. If you have more thoughts or questions, please feel free to share.
> > >
> > > Warm regards,
> > >
> > > Authors

---

### Official Review · Reviewer_48tf · 2023-07-07

**Soundness:** 3 good
**Presentation:** 3 good
**Contribution:** 2 fair
**Rating:** 6
**Confidence:** 4

**Summary:**

In this paper, authors propose a disentanglement framework for speaker recognition that extract speaker representations from the mixture of speaker traits and content. The proposed framework utilizes three Gaussian inference layers to decompose the static and dynamic speech components. Additionally, a self-supervision method is also proposed to help disentangle content without relying on text labels. Based on experimental results, the authors claim that the proposed method achieves remarkable performance on the VoxCeleb and SITW datasets, which demonstrates the effectiveness of the framework.

**Strengths:**

1. The authors conduct solid experiments and the proposed method shows good performance on VoxCeleb and SITW.

2. This paper shows detailed experimental results and a comprehensive comparison with several solid baselines.

3. The paper is well-written and easy to follow.

**Weaknesses:**

1. The proposed RecXi model with self-supervision speaker preserving loss generative very limited improvement compared to the Xi baseline system.

**Questions:**

I was wondering if authors evaluate the proposed method on some more challenging datasets, such as VOiCES?

**Limitations:**

The authors analyze the limitations in Section 6.

---

> ### Author Rebuttal · Authors · 2023-08-08
>
> Dear Reviewer,
>
> Thank you for taking the time to review our work and providing us with your valuable suggestions and insightful comments. We have carefully considered your feedback, and we would like to address your concerns as follows:
>
> **Weakness:** Thanks for pointing this out. We use the best baseline that outperforms those reported in the literature [38, 45, 36, 43, 87, 16, 82, 96, 46, 63, 70, 84] on the same dataset, and we even further enhanced the baseline with the proposed tResNet encoder. The comparison of the baselines and SOTA systems is reported on Page 7, Table 1. In addition, the xi-vector baseline we compared to is also much stronger than that in the original paper [38], on the same dataset (SITW dataset). Since the EER is already very small (below 2%), making further improvements of more than 10% is quite challenging.  As shown in Table 2 and analysis in line 315 in Section 5, by comparing system #10 and #11, "the proposed RecXi achieves average EER/minDCF reductions of 12.15%/10.66% over the Xi baseline."
>
> We apologize if this aspect was not clearly emphasized in the paper. We will add the following sentences to line 303 for clarification: "We observed that the baseline system, comprising a tResNet and Xi combination, is powerful and achieves state-of-the-art (SOTA) performance. This also renders further performance improvements more difficult. However, the proposed disentangling method, RecXi, exhibits even more outstanding performance."
>
>
> **Question:** Thanks for this good suggestion. The reason that we didn’t use VOiCES dataset is that this dataset was specially designed for single channel far-field speech under noisy conditions, which is not the focus of this paper.
>
> And since we already include the easy set (Vox1-O), the hard cases (Vox1-H), the large set (Vox1-E) and a cross domain test set (SITW), the experimental results would be sufficient to support our claims.  But your suggestion is very valuable, we will take note of this and add VOiCES and CNCeleb datasets in future work.
>
> We sincerely appreciate the reviewer for offering a multitude of valuable suggestions. We eagerly anticipate engaging in further discussions during the next phase.

---

> > ### Comment · Reviewer_48tf · 2023-08-21
> >
> > I'd like to thank the authors for their responses and feedback. The explanation of the limited performance improvement makes sense. I will reconsider the review.

---

> > > ### Author Response · Authors · 2023-08-21
> > >
> > > Dear reviewer,
> > >
> > > We greatly appreciate you taking the time to review our rebuttal and for reconsidering the rating. Your valuable feedback has played a crucial role in improving our work. We will include the content of the above rebuttal in our final manuscript.
> > >
> > > Warm regards,
> > >
> > > Authors

---

### Official Review · Reviewer_6npJ · 2023-07-07

**Soundness:** 3 good
**Presentation:** 2 fair
**Contribution:** 2 fair
**Rating:** 6
**Confidence:** 3

**Summary:**

This paper proposes an improvement over Xi-vector speaker recognition algorithm. They propose an iterative 3 layer Gaussian inference mechanism to disentangle the content information from speaker information. The first layer gets precursor speaker representation similar to Xi-vector framework. The second layer uses the encoder embeddings and embeddings of first layer to disentangle the content information. The third layer finally gets the disentangled speaker representation by subtracting the second layer embeddings from encoder embeddings.
They also take inspiration from knowledge distillation framework and use a similarity preserving loss that preserves the similarity between third layer embeddings and the difference between first and second layer embedding. This seems to improve performance in their ablation studies. They compare their model with various baselines that do not disentangle non-speaker info from speaker and that do not use the similarity preserving loss show an improvement in performance.

**Strengths:**

A key contribution of the paper is improving Xi-vector framework to disentangle content info from speaker info. The iterative mechanism to achieve that is well motivated. The proposed similarity preserving loss designed further leads to performance improvement. Another minor contribution of the paper is changing the ResNet34 backbone to a tResNet34 backbone by changing the stride strategies based on the improvement seen in preliminary experiments. That could be a reason why the results reported here with Xi-vector with tResNet34 backbone is better than what is reported in original Xi-vector paper for SITW-eval set leading to a stronger baseline.


**Weaknesses:**

1. The paper did not compare their disentanglement method with any other disentanglement method pre-exisitng in literature. As mentioned in their Related Work section most other disentanglement methods use ASR labels and they might perform better than this method which does not use ASR labels. But since disentanglement is a major contribution of the paper it would have been interesting to compare them in terms of accuracy, training time and model parameters used. Also an approach based on [1, 2] could also have been explored to learn nuisance free disentangled representations for speaker recognition which does not use ASR labels.
2. Another major contribution is the self-supervised similarity preserving loss. However, authors do not compare with other methods pre-existing in literature that use contrastive learning as mentioned in their related work section.
3. Utility of module "G" is not shown in the paper (as defined in "Frame-wise Content Aware Transition Model"). No ablation studies are provided as to if it is required at all. No experiments were done to show the benefit of using multiple G matrices over just one matrix. Without this module the Rec-Xi layer is same as a Xi layer and hence it is important to show its utility and what improvement we get by modifying the baseline Xi layer.

[1] Liu et al. "Towards Learning Nuisance-Free Representations of Speech", ICASSP 2018
[2] Mun et al. "Disentangled Speaker Representation Learning via Mutual Information Minimization"


**Questions:**

1. Authors do one round of content disentanglement in 2nd layer before getting the final speaker representation in 3rd layer. Theoretically, they can keep disentangling content and speaker info for more than 1 iteration. Did authors do any experiment if further rounds of content/speaker disentanglement leads to any performance improvement?
2. How much is the increase in training time when RecXi is used compared with Xi due to its iterative nature?
3. Authors try using just the "phi" and "phi" concatenated with "phi_l" embedding for speaker recognition in their ablation studies Table 3. Did they try using just the "phi_l" vector defined in eqn 24 for speaker prediction to quantify what information it carries?
4. What about adding an extra loss term that uses the precursor embedding to predict speaker info similar to Xi-vector algorithm. One can also concatenate it with the third layer's embedding to get final prediction. Did the authors try any such experiments to see if it can provide additional benefit?
5. Computation of w_t,N in equation 11 is not clear to me. Did the author convert rho_t to a N dimensional vector using the non-linear operation f and take softmax along the N dimensions of the resulting vector to get the vector w_t?

**Limitations:**

Authors discuss limitation and broader impacts of the paper.

---

> ### Author Rebuttal · Authors · 2023-08-08
>
> Dear Reviewer:
>
>
> Thanks for reviewing our work and providing valuable and insightful suggestions. **More experiments are reported in the PDF file in the ‘Global’ rebuttal, as per NeurIPS guideline.** Please let us know if you have any difficulty accessing the file. **Due to the word limit of the rebuttal (6000 characters), we have simplified some of the content below.** We eagerly look forward to further discussions with the reviewer. And here is our rebuttal to address your concerns.
>
> **Weakness 1 & 2**:
>
> **1). Pretrained ASR models**: Thanks for pointing this out. As mentioned in Section I, line 40, "ASR models are typically one or two orders of magnitude larger than the speaker recognition model".  This issue has led to limited adoption of pre-trained ASR models in SOTA methods. A related work [1] published in 2019 utilizes this method. A comparison is presented in Table 1 in the PDF. Notably, our proposed method (#9) outperforms the approach (#New1) from [1].
>
> In addition, we find a recent  work (which became available after our submission)  that utilizes a pre-trained ASR model to initialize the speaker verification model [2]. Due to word limitations, a detailed analysis is presented in the global rebuttal. In summary, our approach achieves comparable performance to the model in [2] while also being **55.5% smaller than its smallest model.**
>
> **2). Text labels**. As mentioned in our paper, acquiring text labels for large-scale speaker verification datasets, such as the VoxCeleb dataset with over 1 million samples, is very expensive. Therefore, to the best of our knowledge, there is no suitable system to compare with.
>
> **3). Contrastive learning.** We sincerely appreciate this valuable comment provided.
> As presented in Table 1, we compare the proposed RecXi and the recently reported SOTA contrastive learning system [3], both evaluated on the same dataset. Notably, even though the models in [3] incorporate extra visual information beyond speech, our proposed RecXi consistently demonstrates substantial superiority over the system detailed in [3] across all three test sets.
>
> **4). Nuisance free disentangled representations**: We appreciate this good point. The reconstruction is required in [4], while as we mentioned in Section II, line 117, the self-supervision is designed to be simple yet effective and avoid extra signal re-contractors. In [5], device labels are used to train the device classifier, whereas in our work, text labels are not available for training. We briefly discuss and cite [5] on Page 2, Section II, line 93, ‘removal of irrelevant information, like **devices**... with corresponding labels [**54**]’.
> These two papers are insightful and serve as valuable references for our related work discussion. The following sentences will be added in Section II, line 93: "[5] utilizes MI estimators to minimize the mutual information between speaker and device embeddings and reduce the interdependence between embeddings and labels. In [4], nuisance variables, such as gender and accent, are removed from speaker embeddings through the learning of two separate orthogonal representations."
>
> [1] Zhou et al. "CNN with Phonetic Attention for Text-Independent Speaker Verification", IEEE ASRU 2019.
>
> [2] Cai et al. "Pretraining Conformer with ASR for Speaker Verification", IEEE ICASSP 2023.
>
> [3] Tao et al. "Self-Supervised Training of Speaker Encoder With Multi-Modal Diverse Positive Pairs", IEEE TASLP 2023.
>
> [4] Liu et al. "Towards Learning Nuisance-Free Representations of Speech", IEEE ICASSP 2018.
>
> [5] Mun et al. "Disentangled Speaker Representation Learning via Mutual Information Minimization", IEEE APSIPA ASC 2022.
>
> **Weakness 3**:  Thanks for this great suggestion. We add additional experiments by replacing the $\mathbf{G_t}$ with single learnable matrix or identity matrix following that in xi-vector, while maintaining the three-layer design with the self-supervision loss. The results are presented in Table 3 in the PDF. A detailed analysis is presented in the global rebuttal. In summary, the results validate the effectiveness of $\mathbf{G_t}$.
>
> **Question 1**: Yes. We agree with the reviewer. While the idea of stacking layer 2 and layer 3 seemed promising, we believe that the benefits of additional layer stacking diminish rapidly. The current three-layer configuration has yielded concrete results, effectively supporting our assumptions and claims.
> We reserve this further investigation as a future work.
>
> **Question 2**: Thanks for raising this question. The training process is 25% slower because of the recursive. However, the increase in Floating Point Operations (FLOPs) is only 13%. It is worth noting that further optimization, similar to the LSTM, could be implemented through coding improvements. Additionally, for larger models, the encoder component consumes a significant portion of the computation time, and the impact of RecXi's computational overhead is expected to diminish in such cases.
>
> **Question 3**: Thanks for this suggestion. We add the ablation study in Table 2 in the PDF. The results clearly show that $\tilde{\phi}_l$ carries speaker information and demonstrates strong discriminative capabilities. A detailed analysis is presented in the global rebuttal.
>
> **Question 4**: Our preliminary results indicate that there is no obvious improvement obtained by adding the loss function for the precursor embedding. Our explanation for this outcome is that the additional loss could complicate gradient propagation and lead to confusion for the model's learning objectives in layer 1.
>
> **Question 5**: Yes, the reviewer’s understanding is correct. We will add Equation 1 in PDF to make it clear to readers.
>
> We express our sincere gratitude to the reviewer for providing numerous valuable suggestions. Inspired by these suggestions, we conducted additional experiments, which yielded more evident results that strongly support our claims and demonstrate the success of disentanglement.

---

> > ### Comment · Reviewer_6npJ · 2023-08-17
> >
> > I thank the authors for providing additional experiments comparing their method with other methods for disentangling content info from speaker and self-supervised method. I appreciate the experiment showing the benefit of the proposed module G. The training time could still be a bottleneck due to iterative nature of the architecture. But I believe the efficacy shown in disentangling content information without using heavy weight ASR models outweighs that drawback. I raise my review to a weak accept.

---

> > > ### Author Response · Authors · 2023-08-17
> > >
> > > Dear reviewer,
> > >
> > > We genuinely appreciate your positive feedback on our rebuttal! Your insights and guidance have been instrumental in shaping the revisions to our paper. We are also deeply thankful for your reconsideration and decision to raise the rating.
> > >
> > > Warm regards,
> > >
> > > Authors

---

### Official Review · Reviewer_ffvX · 2023-07-10

**Soundness:** 3 good
**Presentation:** 2 fair
**Contribution:** 3 good
**Rating:** 7
**Confidence:** 3

**Summary:**

The authors propose RecXi, a Gaussian inference based disentanglement method for speech that itself is based on Xi-Vector. Overall, the methods makes sense with sufficient motivation and extensive experiments. The results are promising, and have further mileage in ASR, voice conversion and TTS.

The paper's main contribution are:
1. Improve Xi-Vector to capture temporal information occurring naturally in speech.
2. A frame-wise content aware transition model to model dynamic aspects of spoken signal.
3. Gaussian inference of 3 layers to disentangle content from speaker.
4. A self-supervision trick to avoid usage of textual supervision.

The authors use a number of strong baselines spanning CNN and TDNN architectures. Across 3 datasets, the RecXi shows strong performance, outperforming the baselines on EER and minDCF.

The authors also do an ablation study to understand the contribution of the proposed methods. While helpful it is not clear what does the system RecXi($\tilde{\phi}$, $\tilde{\phi}_l$) entail (details below in Questions section).

Rebuttal:
I have read the authors' rebuttal, and they have clarified the questions I had. My question was mainly due to the presentation of the equation, and the authors have taken the feedback and agreed to update the notation they have used in equations.

**Strengths:**

1. Well-motivated methods to the central research question of disentanglement of content and speaker.
2. Sound modeling approach with extensive experiments.
3. Good ablation study of the various components of the modeling technique.
4. Has more mileage for work in ASR, TTS and Voice conversion.

**Weaknesses:**

1. The authors claim that disentangling speaker and content information is sufficient for getting accurate speaker representation. However, the effect of prosody is not clearly examined in this work.

2. Reproducibility since it's not clear if the code/model are shared or will be shared in future.

**Questions:**

The concept of $\tilde{\phi}\_l$ is introduced in section 3.3; it is used for computing $l\_{ssl}$. So, what exactly does RecXi($\tilde{\phi}$, $\tilde{\phi}\_l$) entail when $l\_{ssl}$ is not used?

**Limitations:**

As noted by the authors, the self-supervised loss makes intuitive sense but needs more investigation in terms of the loss function itself as well as some probing about why it works. Further, the number of transition models in layer 2 of Gaussian inference is a hyper-parameter whose interpretation is not clear is heavily dependent on experimentation.

---

> ### Author Rebuttal · Authors · 2023-08-06
>
> Dear reviewer,
>
> Thank you for reviewing our work and sharing the valuable suggestions and generous comments! Following is our rebuttal to address the concerns.
>
> **Weakness 1**: Thanks for pointing this out. It is true that we do not consider explicitly prosody features  in our approach. As observed in most SOTA speaker embedding representations, such as the x-vector and xi-vector, the inclusion of prosody remains ambiguous. The empirical experimental results in this paper demonstrate that focusing on the static component of speech, which represents the speaker's characteristics determined by the vocal tract's shape, proves adequate for obtaining accurate speaker embeddings. Nonetheless, an important avenue for future investigation lies in the decomposing prosody from the dynamic components. We express our gratitude for this valuable suggestion.
>
> **Weakness 2**: Thank you for pointing this out. We fully acknowledge the significance of sharing code and pre-trained models to benefit the community and attract more attention. Due to the internal approval procedure in our organizations, it may take a longer time for the code to be available. Alternatively, while we work towards the official release, we will offer help and guidance for any third-party interested in reproducing, particularly for academic purposes.
>
>
> **Question**: We appreciate your question. In our work, $\ella$ denotes the loss function, which  includes components such as $\ella_{cls}$ and $\ella_{ssp}$. The $\mathcal{l}$ in $\tilde{\phi_{l}}$ indicates that it is obtained by a linear operation, differing from $\tilde{\phi}$ in $layer$ 3 which is disentangled by Gaussian inference, as mentioned in Page 6, Section 3.3, line 245. Based on your insightful question, we observe that $\ella$ and $l$ are similar. To avoid any potential misunderstandings, we will replace all the $\tilde{\phi_{l}}$ with $\tilde{\phi_{lin}}$, while referring to the loss function as $\mathcal{L}$.
>
> Sincerely thanks for your valuable suggestions which inspire us and help us to revise this work. We hope our rebuttal adequately addresses your concerns. If anything is still unclear, please feel free to inquire during the discussion phase. We eagerly anticipate further communication.

---

> > ### Comment · Reviewer_ffvX · 2023-08-17
> >
> > Thank you for your response.
> >
> > It's correct that speaker embeddings obtained from static components are effective, and it's reasonable to take on investigating decomposition of prosody from dynamic components as future work.
> >
> > It's great that you're offering help till your internal approvals go through. However, please indicate that you intend to share code and models in near future.
> >
> > Re: question about loss function, I am not able to follow your explanation possibly because Latex notations aren't rendering correctly. Do you mind using plaintext as much as possible to rewrite the response to my question? Thanks!

---

> > > ### Author Response · Authors · 2023-08-18
> > >
> > > Dear reviewer,
> > >
> > > We appreciate your thoughtful response to our rebuttal. Please allow us to address your points:
> > >
> > > 1. We are in agreement with your perspective, and we extend our sincere appreciation for this insightful suggestion. It presents a valuable avenue for future exploration.
> > >
> > > 2. Thanks for your suggestion. We will add this sentence to the paper: "The code and the pretrained models will be made available with third-party re-implementation."
> > >
> > >     The internal approval may take a longer time. In any case, we will have the third-party re-implementation available.
> > >
> > > 3. We apologize that our previous response wasn't sufficiently clear in conveying our points. To ensure clarity, we provide a more comprehensive explanation in both LaTeX and plaintext formats:
> > >
> > > ---
> > > (LaTeX version)
> > >
> > > In our work, the symbol of $\ell$ denotes the loss functions, such as $\ell_{ssp}$ and $\ell_{cls}$ in Eq. 25 and 26.
> > >
> > > The usage of $\mathcal{l}$ in $\tilde{\phi_l}$ indicates that $\tilde{\phi_l}$ is obtained through a linear operation (Eq. 24).
> > > This choice of notation differentiates $\tilde{\phi_l}$ from $\tilde{\phi}$. Specifically, $\tilde{\phi_l}$ denotes the speaker representation resulting from the linear operation, while $\tilde{\phi}$ is the speaker representation disentangled by the Gaussian inference in layer 3, as discussed in Page 6, Section 3.3, line 245.  **Therefore, actually, $\ell$ and $\mathcal{l}$ have different meanings in our work**.
> > >
> > > Your inquiry has brought to light the potential confusion arising from the similarity of these symbols. To avoid any potential misunderstandings, we will replace all the $\tilde{\phi_l}$ with $\tilde{\phi_{\mathrm{lin}}}$, and also change the loss function symbol from $\ell$ to $\mathcal{L}$.
> > >
> > > On the other hand, for RecXi($\tilde{\phi}$, $\tilde{\phi_l}$), when the $\ell_{ssp}$  is missing, $\tilde{\phi_l}$ is trained by the classification loss $\ell_{cls}$. It is accomplished by concatenating $\tilde{\phi_l}$ and $\tilde{\phi}$ (Line 290), and then passing this concatenation to the decoder. The results show that the $\ell_{ssp}$ loss helps in the modeling of speech dynamics.
> > >
> > > ---
> > > (Plaintext version)
> > >
> > > In our work, the symbol of \ell denotes the loss functions, such as \ell_{ssp} and \ell_{cls} in Eq. 25 and 26.
> > >
> > > The usage of \mathcal{l} in \tilde{\phi_l} indicates that \tilde{\phi_l} is obtained through a linear operation (Eq. 24).
> > > This choice of notation differentiates \tilde{\phi_l} from \tilde{\phi}. Specifically, \tilde{\phi_l} denotes the speaker representation resulting from the linear operation, while \tilde{\phi} is the speaker representation disentangled by the Gaussian inference in layer 3, as discussed in Page 6, Section 3.3, line 245. **Therefore, actually, \ell and \mathcal{l} have different meanings in our work.**
> > >
> > > Your inquiry has brought to light the potential confusion arising from the similarity of these symbols. To avoid any potential misunderstandings, we will replace all the \tilde{\phi_l} with \tilde{\phi_{\mathrm{lin}}}, and also change the loss function symbol from \ell  to \mathcal{L}.
> > >
> > > On the other hand, for RecXi(\tilde{\phi}, \tilde{\phi_l}), when the \ell_{ssp} is missing, \tilde{\phi_l} is trained by the classification loss \ell_{cls}. It is accomplished by concatenating \tilde{\phi_l} and \tilde{\phi} (Line 290), and then passing this concatenation to the decoder. The results show that the \ell_{ssp} loss helps in the modeling of speech dynamics.
> > >
> > > ---
> > > Once again, we sincerely apologize for any confusion caused by the earlier explanation. We trust that this detailed clarification will be beneficial. Please do not hesitate to reach out if you have any further queries.

---

### Author Rebuttal · Authors · 2023-08-08

Dear all reviewers,

We greatly appreciate the insightful suggestions and valuable comments from each reviewer. These have been immensely helpful and enlightening for refining this paper.

We have included a figure and several experimental results to aid our rebuttal. **Following the NeurIPS guidelines for this year, we have placed these tables and figures in the attached PDF file of this global rebuttal.** Please let us know if you are unable to access this PDF. The following is the analysis of the content in the PDF.

**Figure 1: t-SNE visualization of speaker discriminative ability for ablation study.** (related to reviewer P6Zh, Question 1)

**Analysis**: The speakers used in Figure 1 are taken from the test set of VoxCeleb.The visualization depicts speakers from the test set of VoxCeleb, where the first 200 speakers (indexed from id10001 to 10200) are included, each with 20 utterances. From the figure, it is evident that in layer 3, the disentangled speaker representations ($\tilde{\phi}$) are highly discriminative for speakers. Moreover, benefiting from the self-supervision loss, the speaker representation $\tilde{\phi_{l}}$ acquired through the linear operation (Eq. 24) also exhibits notable speaker discriminative ability, comparable to that derived from layer 3 ($\tilde{\phi}$). However, for layer 2 ($\rho$), as its main objective is to disentangle content information, it lacks the discriminative ability observed in layer 3 ($\tilde{\phi}$). These observations align with those in Table 2 of the attached PDF and strongly support our claims.


**Table 1: A comparison between proposed RecXi with SOTA methods using pre-trained ASR models or contrastive learning.**  (related to reviewer 6npJ, Weakness 1 & 2; reviewer krxU, Weakness 2)

**Analysis**: Thank reviewers for kind understanding that we didn't compare the ASR model-based systems because of their significantly larger model size compared to the speaker recognition model.
This issue has led to limited adoption of pre-trained ASR models in SOTA methods. A related work [1] published in 2019 utilizes this method. A comparison is presented in Table 1. Notably, our proposed method (#9) outperforms the approach (#New1) from [1].

In addition, we find a recent work (which became available after our submission)  that utilizes a pre-trained ASR model to initialize the speaker verification model [2]. In Table 1, we observe that RecXi outperforms the model without pre-training in [2] (system #9 vs #New2). Additionally, the proposed method achieves similar performance with the model utilizing ASR pre-training in [2] (system #9 vs #New3). Our proposed method offers a significant advantage: it achieves competitive performance without requiring pre-training of an ASR model. Additionally, our method is approximately 55.5% smaller than the smallest model in [2], highlighting the efficiency and effectiveness of the proposed RecXi.

We also compare the proposed RecXi and the recently reported SOTA contrastive learning system, both evaluated on the same dataset. Notably, even though the models in [3] incorporate extra visual information beyond speech, our proposed RecXi consistently demonstrates substantial superiority over the system detailed in [3] across all three test sets.

[1] Zhou et al. "CNN with Phonetic Attention for Text-Independent Speaker Verification", IEEE ASRU 2019.

[2] Cai et al. "Pretraining Conformer with ASR for Speaker Verification", IEEE ICASSP 2023.

[3] Tao et al. "Self-Supervised Training of Speaker Encoder With Multi-Modal Diverse Positive Pairs", IEEE TASLP 2023.

**Table 2: Ablation study using different posteriors from the three layers of RecXi.**  (related to reviewer 6npJ, Question 3; reviewer P6Zh, Weakness 1)

**Analysis**: The results clearly show that both $\tilde{\phi}$ and $\tilde{\phi_{l}}$ carries speaker information and demonstrates strong discriminative capabilities. Additionally, reporting on layer 2 further supports our claims, confirming its role in representing content while effectively removing speaker-related information. The EER being close to the maximum value, 50%, indicates that layer 2 does not contain any speaker-related information and does not exhibit any speaker discriminative ability. The precursor speaker representation ${\phi}$ also exhibits fine speaker discriminative ability, but it is slightly inferior to $\tilde{\phi}$. We express our deep gratitude to the reviewers for inspiring us to conduct this ablation study, which unmistakably demonstrates the success of our disentanglement approach.

**Table 3: Verification for the effectiveness of $\mathbf{G}_t$** (related to reviewer 6npJ, Question 5)

**Analysis**: To verify the effectiveness of $\mathbf{G}_t$, we conduct experiments by replacing the $\mathbf{G_t}$ with single learnable matrix (#New10) or the identity matrix (#New11) following that used in the xi-vector, while maintaining the three-layer design with the self-supervision loss. Based on the results, it is evident that using an identity matrix or a single learnable matrix leads to poor performance. This clearly demonstrates that if the second layer lacks the capacity to model dynamic patterns, the model becomes confused. This, in turn, has a significant impact on layer 3's ability to obtain accurate speaker embeddings, as it relies on removing the content information provided by layer 2. Above observations further validate the effectiveness of the proposed $\mathbf{G_t}$. Furthermore, when N is set to 6 (#New9), the model achieves performance that is better than xi-vector baseline (#4 on Page 7) and is close to N=16 (#5) but slightly worse. This demonstrates the necessity of our proposed $\mathbf{G_t}$ in facilitating the capability of layer 2 to model dynamic components.

**Equation 1**  (related to reviewer 6npJ, Weakness 3)

This equation is utilized to provide a clearer demonstration of the derivation of $\mathbf{G}_t$, and it will be included in the paper.

---

### Decision · Program_Chairs · 2023-09-21

**Decision:**

Accept (poster)

**Comment:**

The authors introduce an innovative three-layer Gaussian inference mechanism to disentangle content information from speaker characteristics. The first layer creates a precursor speaker representation akin to the Xi-vector framework. The second layer employs the encoder embeddings along with the first layer's embeddings to disentangle content information. Lastly, the third layer yields a fully disentangled speaker representation by subtracting the second layer's embeddings from the encoder embeddings.

A primary critique raised by reviewers was the paper's need for comparing with more existing disentanglement methodologies. Specifically, there was a noticeable absence of comparisons to methods employing ASR and contrastive learning. In their rebuttal, the authors address this concern by including comparisons with ASR-based methods published in 2019 and a recent work that became available post-submission. They also add more comparisons using contrastive learning methods. Impressively, the proposed approach demonstrates competitive performance without necessitating the pre-training of an ASR model. I believe this effectively addresses the concerns initially raised by the reviewers.